# PRIOR-INFORMED FLOW MATCHING FOR GRAPH RECONSTRUCTION

## ABSTRACT

We introduce *Prior-Informed Flow Matching (PIFM)*, a conditional flow model for graph reconstruction. Reconstructing graphs from partial observations remains a key challenge; classical embedding methods often lack global consistency, while modern generative models struggle to incorporate structural priors. PIFM bridges this gap by integrating embedding-based priors with continuous-time flow matching. Grounded in a permutation equivariant version of the distortion-perception theory, our method first uses a prior, such as graphons or GraphSAGE/node2vec, to form an informed initial estimate of the adjacency matrix based on local information. It then applies rectified flow matching to refine this estimate, transporting it toward the true distribution of clean graphs and learning a global coupling. Experiments on different datasets demonstrate that PIFM consistently enhances classical embeddings, outperforming them and state-of-the-art generative baselines in reconstruction accuracy.

## 1 INTRODUCTION

Graph generative models have seen remarkable progress in recent years, enabling the synthesis of realistic graph structures in domains such as drug design (Yang et al., 2024) and social networks (Grover et al., 2019). In particular, diffusion-based (Niu et al., 2020; Jo et al., 2022; Vignac et al., 2023) and flow-based (Qin et al., 2025; Eijkelboom et al., 2024) approaches have emerged as state-of-the-art. While these models excel at *unconditional* generation and property-controlled generation, their application to inverse problems, and in particular, the reconstruction of a graph from partial observations, remains a fundamental open problem.

Graph reconstruction is a long-standing problem, traditionally framed as a link prediction task. Early transductive methods, such as Node2Vec (Grover & Leskovec, 2016; Perozzi et al., 2014), model edges independently and fail to capture global structural information. While inductive methods (Zhang & Chen, 2018) like GraphSAGE (Hamilton et al., 2017) can capture expressive local patterns, they still lack a global perspective on the graph's structure. Conversely, recent generative models adapted from image inpainting (Vignac et al., 2023; Trivedi et al., 2024) or guided by posterior sampling (Sharma et al., 2024; Tenorio et al., 2025) can produce plausible completions but are not optimized for the faithful recovery of the ground truth. This leaves open a critical gap: classical and heuristic-based methods are local, while modern solvers are not designed for exact reconstruction.

In this work, we bridge this gap by introducing **Prior-Informed Flow Matching (PIFM)**, a flow-based model designed for high-fidelity graph reconstruction. We reformulate the problem through the lens of the perception-distortion trade-off (Blau & Michaeli, 2018), which postulates that an optimal estimator can be constructed in two stages (Freirich et al., 2021; Ohayon et al., 2025): $(i)$ predicting the Minimum Mean Squared Error (MMSE) estimator from local information, and $(ii)$ learning an optimal transport map from this initial estimate to the ground-truth graph distribution.

Our method approximates this two-step solution. For $(i)$, we represent the posterior mean as the expected value of a Bernoulli latent variable model with unknown probabilities, where the latent structure is estimated using inductive (dataset-informed, such as graphons (Lovász, 2012) and GraphSAGE (Hamilton et al., 2017)) or transductive estimators (instance-specific, such as node2vec (Grover & Leskovec, 2016)). Then, for $(ii)$, we approximate the optimal transport step using a rectified flow model (Liu et al., 2023; Albergo et al., 2023), which maps the posterior mean to

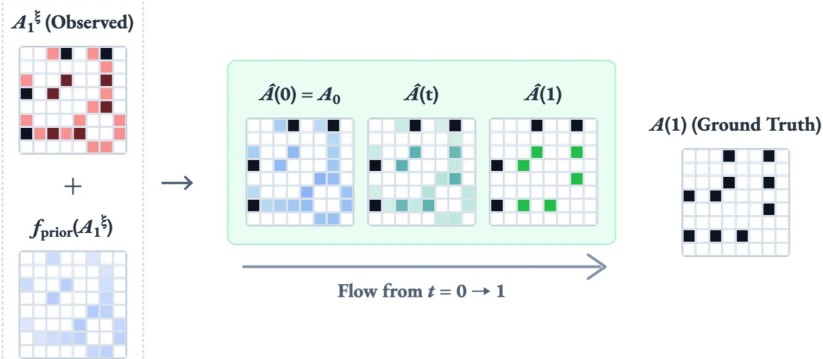

Figure 1: Overview of the **Prior-Informed Flow Matching (PIFM)** graph reconstruction framework. Starting from a partially observed adjacency matrix $\mathbf{A}_1^\xi = \xi \odot \mathbf{A}$, where $\xi$ denotes a mask, we form an initialization $\mathbf{A}_0$ by combining the observed entries with prior predictions $f_{\text{prior}}(\mathbf{A}_1^\xi)$ obtained with an element-wise predictor. In dark red we denote the true edges that are masked, while in light red those masked position that do not have an edge between nodes. A rectified flow then interpolates linearly from $\mathbf{A}_0$ to the ground-truth graph $\mathbf{A}_1 = \mathbf{A}$, learning global structural information from a coupling of all the edges. The intermediate states $\mathbf{A}_t$ improve on the prior-informed initialization, enabling recovery of the missing edges.

the distribution of clean graphs (see Fig. 1). Importantly, our architectures are permutation-equivariant, ensuring a permutation-invariant parameterization of the posterior distribution.

We validate the advantages of PIFM through experiments on datasets with diverse characteristics, including dense and sparse graphs. Our results show that PIFM effectively integrates structural priors with flow-based modeling and can be interpreted as a form of graph inpainting, where missing edges are inferred through a learned interpolation process.

Our contributions are as follows:

- We introduce a novel formulation for graph reconstruction based on a permutation-equivariant distortion-perception trade-off.

- We propose PIFM, a new estimator based on flow matching that defines a prior-informed source distribution using embeddings from latent graph models. PIFM enhances these initial embeddings by learning a global structural coupling.

- We empirically validate our approach on link prediction, and two *blind* versions termed expansion (recover the missing edges) and denoising (removing the spurious edges), showing that PIFM significantly improves the reconstruction performance of predictors that rely solely on local information.

## 2 RELATED WORKS

**Flow/Diffusion models on graphs.** Diffusion and flow-based graph generative models have shown impressive performance in recent years. Early models, namely EDP-GNN (Niu et al., 2020) and GDSS (Jo et al., 2022), employ score-based *continuous* diffusion over a relaxation of the graph structure. However, given that graphs are inherently discrete, subsequent work has explored discrete diffusion processes (Austin et al., 2021). Models like DiGress (Vignac et al., 2023) demonstrated the effectiveness of this approach, which has been further advanced by discrete flow-based models like DeFoG (Qin et al., 2025) and variational approaches like CatFlow (Eijkelboom et al., 2024). A common point of these models is their reliance on a simple source distribution, such as Gaussian (continuous) or uniform (discrete) noise. While effective for unconstrained generation, recent work on image-based inverse problems demonstrates the advantages of learning a data-dependent flow, using a prior-informed source distribution (Albergo et al., 2023; Delbracio & Milanfar, 2024; Ohayon et al., 2025).

**Graph topology inference via flow/diffusion-based solvers.** Graph topology inference – the task of recovering hidden edges from a partially observed graph – is a long-standing inverse problem (Segarra et al., 2017; Dong et al., 2016). Several methods adapt diffusion for constrained graph generation, which is related to but distinct from topology inference. DiGress (Vignac et al., 2023) introduced an inpainting mechanism, inspired by Repaint (Lugmayr et al., 2022), to generate graph structures consistent with a partial observation. Similarly, in Trivedi et al. (2024) a similar mechanism is used for completing partially observed graphs. PRODIGY (Sharma et al., 2024) enforces hard constraints by projecting the graph estimate onto a feasible set at each sampling step. More recently, GGDiff (Tenorio et al., 2025) incorporates a guidance mechanism as a flexible alternative to inpainting. However, all these methods are designed for *constrained generation* (e.g., molecule generation with a given scaffold) rather than *recovering masked edges from a partially observed graph*. Hence, to the best of our knowledge, designing a diffusion-based model explicitly for graph topology inference remains an open problem.

## 3 BACKGROUND

We represent an undirected graph $\mathcal{G}_0 = \{\mathcal{V}, \mathcal{E}\}$, where $\mathcal{V}$ denotes the nodes and $\mathcal{E}$ the edges, by its binary symmetric adjacency matrix $\mathbf{A}_0 \in \mathbb{R}^{N \times N}$.

**Continuous flow matching for graph generation.** Flow matching (Albergo et al., 2023; Lipman et al., 2023) is a family of generative models that defines a continuous-time transport map from samples $\mathbf{A}_0$ drawn from a source distribution $p_0$ to samples $\mathbf{A}_1$ from a target distribution $p_1$. It is governed by the ODE

$$d\mathbf{A}_t = v(\mathbf{A}_t, t)\, dt, \tag{1}$$

where $v(\cdot, t)$ is a velocity field and $\mathbf{A}_t$ denotes a forward process, also known as stochastic interpolant, for $t \in [0, 1]$. Typically, $p_0$ is a tractable distribution (e.g., a Gaussian distribution), while $p_1$ corresponds to the data distribution. To generate new samples, one must specify both $\mathbf{A}_t$ and $v$. A common choice for the forward process is $\mathbf{A}_t = \alpha_t \mathbf{A}_0 + \beta_t \mathbf{A}_1$, where $\alpha_t$ and $\beta_t$ are differentiable functions such that $\alpha_0 = 1$, $\beta_0 = 0$ and $\alpha_1 = 0$, $\beta_1 = 1$. Differentiating this path gives a velocity $v(\mathbf{A}_t, t) = \dot{\alpha}_t \mathbf{A}_0 + \dot{\beta}_t \mathbf{A}_1$. Despite its closed-form, this expression depends explicitly on $\mathbf{A}_1$, making it impractical since the target is unknown at inference/sampling. To circumvent this, we instead consider $v(\mathbf{A}_t, t) = \mathbb{E}_{\mathbf{A}_0, \mathbf{A}_1}[\dot{\alpha}_t \mathbf{A}_1 + \dot{\beta}_t \mathbf{A}_0 \mid \mathbf{A}_t]$, the conditional expectation of the velocity given $\mathbf{A}_t$ (Albergo et al., 2023), which is then approximated with a neural network $v_\theta$. The network is trained using a mean squared error loss:

$$\mathbb{E}_{t, \mathbf{A}_0, \mathbf{A}_1} \left[ \left\| v_\theta(\mathbf{A}_t, t) - (\dot{\alpha}_t \mathbf{A}_1 + \dot{\beta}_t \mathbf{A}_0) \right\|_2^2 \right]. \tag{2}$$

In particular, this formulation does not require $\mathbf{A}_0$ and $\mathbf{A}_1$ to be independent; in fact, they might be sampled from a joint distribution, allowing for richer transport plans in cases where paired data is available. This has been exploited to solve inverse problems on images (Ohayon et al., 2025; Albergo et al., 2024; Delbracio & Milanfar, 2024), and is directly related to our proposed method, as described later.

Throughout this work, we consider the *rectified flow* case (Liu et al., 2023), where $\alpha_t = 1 - t$ and $\beta_t = t$. As shown in Tong et al. (2024), the velocity field associated with this linear path approximates the optimal transport vector field when the joint distribution $p(\mathbf{A}_0, \mathbf{A}_1)$ closely resembles the optimal coupling between the marginals $p(\mathbf{A}_0)$ and $p(\mathbf{A}_1)$. We deferred to Appendix C.2 a more detailed background on generative models on graphs beyond continuous flow matching, including diffusion-based models, as well as related works.

## 4 METHOD

In Section 4.1, we introduce the distortion-perception trade-off (Blau & Michaeli, 2018) for graphs. Then, in Section 4.2, we introduce methods for approximating the posterior mean. In Section 4.3, we describe our implementation of the flow model to transport the predicted mean to the ground-truth graphs.

## 4.1 GRAPH TOPOLOGY INFERENCE AS A DISTORTION-PERCEPTION TRADE-OFF

We aim to reconstruct the ground-truth adjacency matrix $\mathbf{A}$ of graph $\mathcal{G}$ from a partially observed version, denoted by $\mathbf{A}^{\mathcal{O}}$. This task can be formalized through the following distortion-perception function:

$$D(P) = \min_{p(\hat{\mathbf{A}}|\mathbf{A}^{\mathcal{O}})} \left\{ \mathbb{E}_{p(\mathbf{A},\hat{\mathbf{A}})}[\|\mathbf{A} - \hat{\mathbf{A}}\|_F^2] \ : \ d(p_{\mathbf{A}}, p_{\hat{\mathbf{A}}}) \leq P \right\}, \tag{3}$$

where $\hat{\mathbf{A}}$ is an estimator of $\mathbf{A}$ given the observation $\mathbf{A}^{\mathcal{O}}$, and $d(p_{\mathbf{A}}, p_{\hat{\mathbf{A}}})$ is a divergence between the distributions $p_{\mathbf{A}}$ (the distribution of clean graphs) and $p_{\hat{\mathbf{A}}}$. Although we adopt MSE as the distortion measure in (3), the formulation is general and supports other distortion metrics between the true and predicted graphs (Blau & Michaeli, 2018). The function in (3) has been extensively studied in the image domain (Freirich et al., 2021), where different values of $P$ correspond to estimators with varying characteristics in terms of average accuracy (distortion), and the degree to which the reconstructed signal looks like the ground truth (perception). We adapt this framework to graphs by estimating the matrix $\mathbf{A}$, and accounting for symmetry constraints due to the permutation invariance of graph representations.

Among all possible values of $P$, the most studied cases are $P = \infty$ and $P = 0$. The former corresponds to the distortion function $D(P = \infty)$, whose solution is the posterior mean estimator $\hat{\mathbf{A}}^* = \mathbb{E}[\mathbf{A} \mid \mathbf{A}^{\mathcal{O}}]$. This estimator minimizes distortion, but does not impose constraints on the distribution of outputs, potentially leading to unrealistic graphs. In contrast, when $P = 0$, the estimator achieves a perfect perceptual reconstruction, meaning that the recovered graph has the same structural properties than the original graph; in this case, $p_{\mathbf{A}} = p_{\hat{\mathbf{A}}}$. As shown in Freirich et al. (2021), the corresponding estimator can be obtained by solving the following optimal transport problem

$$p_{\hat{\mathbf{A}},\hat{\mathbf{A}}^*}^* = \operatorname*{argmin}_{p \in \Pi(p_{\mathbf{A}}, p_{\hat{\mathbf{A}}^*})} \mathbb{E}[\|\hat{\mathbf{A}} - \hat{\mathbf{A}}^*\|_F^2], \tag{4}$$

where $\Pi(p_{\mathbf{A}}, p_{\hat{\mathbf{A}}^*})$ is the set of all joint distributions (couplings) with fixed marginals $p_{\mathbf{A}}$ and $p_{\hat{\mathbf{A}}^*}$. Thus, finding the estimator $\hat{\mathbf{A}}$ associated with $D(0)$ boils down to solving the optimal transport problem between the distribution of clean graphs $p_{\mathbf{A}}$ and of the MMSE estimator $p_{\hat{\mathbf{A}}^*}$. We can approximate this by $(i)$ computing $\hat{\mathbf{A}}^* = \mathbb{E}[\mathbf{A} \mid \mathbf{A}^{\mathcal{O}}]$ given an observation $\mathbf{A}^{\mathcal{O}}$ and $(ii)$ sampling from the conditional distribution $p(\mathbf{A} \mid \hat{\mathbf{A}}^*)$. Intuitively, this approach builds the final prediction by refining the initial guess $\hat{\mathbf{A}}^*$.

While the solution for $P = \infty$ achieves lower MSE, it may produce outputs that deviate from the structural properties of the original data. This mismatch is problematic in settings like conditional molecular generation, where the generated molecule must satisfy strict chemical validity constraints. For such applications, the solution with $P = 0$ is more appropriate, as it guarantees that the generated samples are structurally consistent with the data distribution. Therefore, this work focuses on the case $P = 0$.

**Permutation invariance on graphs: An additional constraint.** Unlike images, graphs lack canonical node ordering, which imposes symmetry constraints on the data distribution and the reconstruction function. First, the ground-truth distribution $p(\mathbf{A})$ is *permutation invariant*, meaning that for any permutation matrix $\mathbf{P}_\pi$ associated with a node relabeling $\pi$, it holds that $p(\mathbf{A}) = p(\mathbf{P}_\pi^\top \mathbf{A} \mathbf{P}_\pi)$. Second, the estimator $\hat{\mathbf{A}} = f(\mathbf{A}^{\mathcal{O}})$ must be *permutation equivariant*, i.e., $f(\mathbf{P}_\pi^\top \mathbf{A}^{\mathcal{O}} \mathbf{P}_\pi) = \mathbf{P}_\pi^\top f(\mathbf{A}^{\mathcal{O}}) \mathbf{P}_\pi$, ensuring that relabeling the input results in a consistently relabeled output. These conditions are necessary for solutions to (3) and (4) to be invariant to the node labeling. We now describe how to implement the solution to (4).

## 4.2 APPROXIMATING THE POSTERIOR MEAN

As discussed in Section 4.1, our goal is to approximate the conditional mean $\mathbb{E}[\mathbf{A} \mid \mathbf{A}^{\mathcal{O}}]$ with a *permutation equivariant* estimator. Before moving to particular parameterizations of the conditional mean, we introduce two assumptions.

**AS 1.** *We assume each edge in $\mathbf{A} \in \{0,1\}^{n \times n}$ follows a Bernoulli distribution whose probabilities depend on latent node variables $\mathbf{z}_1, \ldots, \mathbf{z}_n \in \mathcal{Z}$ such that:*

$$A_{ij} \sim Bernoulli(f(\mathbf{z}_i, \mathbf{z}_j)), \qquad 1 \leq i < j \leq n. \tag{5}$$

*The function $f$ maps pairs of latent variables to edge probabilities, i.e., $f(\mathbf{z}_i, \mathbf{z}_j) = P(A_{ij}|\mathbf{z}_i, \mathbf{z}_j) = p_{ij}$.*

**AS 2.** *We assume that the edges are conditionally independent given the latent structure, i.e., given the latent structure $Z = \{\mathbf{z}_1, \ldots, \mathbf{z}_n\}$, we have:*

$$P(\mathbf{A} \mid Z) = \prod_{1 \leq i < j \leq n} P(A_{ij} \mid \mathbf{z}_i, \mathbf{z}_j). \tag{6}$$

Under these two assumptions, and assuming access to the mapping $\mathbf{z}^{-1} : \mathbf{A} \to \mathcal{Z}$, the posterior mean can be computed element-wise as $\mathbb{E}[\mathbf{A} \mid \mathbf{A}^{\mathcal{O}}] = P(\mathbf{A} \mid \mathbf{z}^{-1}(\mathbf{A}^{\mathcal{O}}))$. We adopt two different type of priors : $(i)$ inductive methods, represented by *graphons* (Lovász, 2012; Avella-Medina et al., 2018), which are bounded, symmetric and measurable functions $\mathcal{W} : [0, 1]^2 \to [0, 1]$, and *GraphSAGE* (Hamilton et al., 2017), a GNN-based estimator and $(ii)$ transductive ones, obtained from *node2vec* (Grover & Leskovec, 2016), which provides an instance-level learned probabilistic model.

**Posterior mean using inductive methods (dataset-informed).** We approximate the posterior mean using two distinct dataset-informed, inductive approaches: graphons and GraphSAGE.

A *graphon*, defined as a symmetric function $\mathcal{W} : [0, 1]^2 \to [0, 1]$, serves as a generative model for a family of graphs:

$$z_i \sim \text{Uniform}[0, 1], \quad i = 1, \ldots, n, \tag{7}$$
$$A_{ij} \sim \text{Bernoulli}(\mathcal{W}(z_i, z_j)), \quad 1 \leq i < j \leq n.$$

Graphons provide a functional representation of exchangeable random graphs where the conditional edge probability is $[\mathbb{E}[\mathbf{A} \mid \mathbf{z}]]_{ij} = \mathcal{W}(z_i, z_j)$. This offers a natural, permutation-equivariant framework for estimating the posterior mean, though it requires access to the inverse mapping $z_i = [\mathbf{z}^{-1}(\mathbf{A}^{\mathcal{O}})]_i$. Since $\mathcal{W}$ is unknown, we estimate it using Scalable Implicit Graphon Learning (SIGL) (Azizpour et al., 2025), which combines a graph neural network (GNN) encoder with an implicit neural representation (INR). SIGL operates in three steps: (1) a GNN-based sorting step to estimate latent node positions $\mathbf{z}$; (2) a histogram approximation of the sorted adjacency matrices; and (3) learning a graphon parameterization $f_\phi$ by minimizing its error against the histograms. A key feature of SIGL is its ability to recover the inverse mapping $\mathbf{z}^{-1}$, making it uniquely suitable for our model (Xia et al., 2023).

As an alternative inductive method, we use *GraphSAGE* (Hamilton et al., 2017). We train the model on the partially observed graphs in the dataset to produce node embeddings $\{\mathbf{z}_i\}_{i=1}^N$. From these embeddings, we train a single logistic predictor on Hadamard edge features $(\mathbf{z}_i \odot \mathbf{z}_j)$ to estimate edge probabilities. The resulting conditional mean is parameterized as $\left[\mathbb{E}[\mathbf{A} \mid \mathbf{A}^{\mathcal{O}}]\right]_{ij} = f_\phi(\mathbf{z}_i \odot \mathbf{z}_j)$.

**Posterior mean using transductive methods (instance-specific).** For a transductive approach, we use *node2vec* to learn an instance-specific embedding and predictor for each graph. Similar to the GraphSAGE method, we first train node2vec on a partially observed graph to obtain node embeddings $\{\mathbf{z}_i\}_{i=1}^N$. However, in contrast to the single predictor used for GraphSAGE, we fit a distinct, *per-graph* logistic link predictor on Hadamard edge features with balanced negative sampling. This yields the same conditional mean parameterization, $\left[\mathbb{E}[\mathbf{A} \mid \mathbf{A}^{\mathcal{O}}]\right]_{ij} = f_\phi(\mathbf{z}_i \odot \mathbf{z}_j)$, but with a predictor $f_\phi(\cdot)$ that is unique to each graph instance. At inference time, this instance-specific model is used to evaluate all masked pairs to compute the posterior mean.

### 4.3 LEARNING THE FLOW MODEL

We now approximate the posterior density $p(\mathbf{A}, \hat{\mathbf{A}}^*)$ by learning a flow model. As explained in Section 3, we need to specify the forward path $\mathbf{A}_t$ and the velocity field $v$. For the former, inspired by Ohayon et al. (2025), we incorporate prior information as the initialization of the forward path; with slight abuse of notation, we denote $f_{\text{prior}}$ as the prediction of the full graph (i.e., $f_{\text{prior}}(\mathbf{A}^{\mathcal{O}}) \triangleq \mathbb{E}[\mathbf{A} \mid \mathbf{A}^{\mathcal{O}}]$). Specifically, we compute the sample $\mathbf{A}_0$ from the source distribution as follows:

$$\mathbf{A}_0 = \xi \odot \mathbf{A} + (1 - \xi) \odot (f_{\text{prior}}(\xi \odot \mathbf{A}) + \boldsymbol{\epsilon}_s), \tag{8}$$

where $\mathbf{A}$ is the ground-truth graph, $\xi$ is the corresponding mask (taking value 1 for the observed pairs of nodes and 0 otherwise), and $f_{\text{prior}}(\xi \odot \mathbf{A})$ is our approximate MMSE estimator for the masked edges. We also add a small amount of noise $\boldsymbol{\epsilon}_s \sim \mathcal{N}(0, \sigma_s^2)$ following Albergo et al. (2024). We define $\mathbf{A}_1 = \mathbf{A}$ for the target distribution.

Regarding the velocity field $v$, we use the architecture from Jo et al. (2022), a GNN-based network that yields a permutation-equivariant parameterization (see Appendix E for details). Combined with the assumptions for the posterior mean parameterization in Section 4.2, this gives a permutation-invariant parameterization of the target density $p(\mathbf{A}_1)$. Since graphs are exchangeable, the target density should not depend on node ordering, making permutation invariance a desirable property. This is formalized in Theorem 1.

**Theorem 1.** *Let the prior estimator* $f_{prior} : \mathbb{R}^{N \times N} \to \mathbb{R}^{N \times N}$ *and the velocity field* $v_\theta : \mathbb{R}^{N \times N} \times [0, 1] \to \mathbb{R}^{N \times N}$ *both be* **permutation-equivariant**. *For any ground-truth graph* $\mathbf{A}_1$ *and mask* $\xi$, *define the flow path from* $t = 0$ *to* $t = 1$ *as:*

$$\mathbf{A}_t = (1 - t)\mathbf{A}_0 + t\mathbf{A}_1, \quad where \quad \mathbf{A}_0 = \xi \odot \mathbf{A}_1 + (1 - \xi) \odot f_{prior}(\xi \odot \mathbf{A}_1).$$

*Then the estimated density for* $\mathbf{A}_1$, *computed as*

$$\log p(\mathbf{A}_1) = \log p(\mathbf{A}_0) - \int_0^1 \text{tr}\left(\frac{\partial v_\theta(\mathbf{A}_t, t)}{\partial \mathbf{A}_t}\right) dt, \tag{9}$$

*is guaranteed to be* **permutation-invariant**.

The proof can be found in Appendix D.1. Since we know that the target probability density should be permutation invariant, Theorem 1 guarantees that we are introducing the right inductive bias by learning a distribution within the family of permutation invariant distributions.

**Final algorithm.** In Alg. 1, we describe our training and sampling algorithms. In essence, PIFM is a general framework that learns a global graph structure to enhance simple, conditionally independent edge-wise priors.

To illustrate what we mean by *learning a global and dependent predictor*, we now describe a toy experiment. Consider a four-node graph $\mathcal{G}$ (see Fig. 2 (a)) where the goal is to predict the diagonal edges under a specific constraint: the only valid outcomes are that both edges are present or both are absent, i.e., $\mathcal{E} = \{[e_{02} = 1, e_{13} = 1], [e_{02} = 0, e_{13} = 0]\}$. Moreover, we assume that the probability of observing the first case is 0.6, while the second one is 0.4.

We first train an edge-wise prior using node2vec, which yields a probability of 0.6 for each diagonal edge. Crucially, because node2vec models each edge prediction independently, this prior is misspecified. A standard predictor based on this prior would always predict $[1, 1]$ if used as conditional mean or, if sampling were to be performed, could generate invalid predictions such as $[1, 0]$.

We then train a flow model using this node2vec prior to construct the initial state $\mathbf{A}_0$ as in (8). After training (see Appendix E for details), we generate 200 samples, illustrated in Fig. 2(b); the proportion of each mode is shown in Fig. 2(c). The results clearly demonstrate that the flow model $(i)$ successfully leverages global information, learning a *probabilistic coupling* between the edges, to generate samples only from the two valid states, and $(ii)$ learns the probability of each mode.

# 5 EXPERIMENTS

## 5.1 SETUP

We evaluate our method on three graph datasets: IMDB-B, PROTEINS, and ENZYMES. Thus, we focus on families of graphs that are diverse to show that our model learns a general predictor. Future work will focus on scaling to larger graphs, such as Cora. Each dataset is split into 85% train, 10% validation, and 5% test graphs, and we evaluate reconstruction quality under two masking levels (10% and 50% of edges, masks generated uniformly at random). The implementation details are provided in Appendix E.

**Evaluation metrics.** Performance is measured exclusively on masked edges. We report both threshold-dependent classification metrics (FPR, FNR) and threshold-independent metrics (ROC-AUC, AP). Threshold-dependent metrics are computed by binarizing predictions at a fixed cutoff of

---

**Algorithm 1** Training and Sampling

   **Training**
1: Sample $\mathbf{A}_1 \sim p(\mathbf{A})$, a mask $\xi$, and time $t \sim U[0, 1]$.
2: Train MMSE estimator: $f_{prior}(\mathbf{A}^{\mathcal{O}})$
3: Compute $\mathbf{A}_0 \triangleq \xi \odot \mathbf{A}_1 + (1 - \xi) \odot \left( f_{\text{prior}}(\mathbf{A}_1^{\mathcal{O}}) + \boldsymbol{\epsilon}_s \right), \quad \boldsymbol{\epsilon}_s \sim \mathcal{N}(0, \sigma_s^2)$
4: Compute $\mathbf{A}_t \triangleq (1 - t)\mathbf{A}_0 + t\mathbf{A}_1$.
5: Train flow model: $\theta^* = \operatorname{argmin}_\theta \mathbb{E}_{\mathbf{A}_1, \mathbf{A}_0, \xi, t} \| v_\theta(\mathbf{A}_t, t) - (\mathbf{A}_1 - \mathbf{A}_0) \|_F^2$
   **Sampling (Reconstruction)**
6: Initialize $\hat{\mathbf{A}} \leftarrow \xi \odot \mathbf{A}_1^{\mathcal{O}} + (1 - \xi) \odot f_{\text{prior}}(\mathbf{A}_1^{\mathcal{O}}) + (1 - \xi) \odot \boldsymbol{\epsilon}_s, \quad \boldsymbol{\epsilon}_s \sim \mathcal{N}(0, \sigma_{\text{samp}}^2)$.
7: **for** $i \leftarrow 0, \dots, K - 1$ **do**
8:      $\hat{\mathbf{A}} \leftarrow \hat{\mathbf{A}} + \frac{1}{K} v_{\theta^*}\left( \hat{\mathbf{A}}, \frac{i}{K} \right)$
9: **end for**
10: Return $\hat{\mathbf{A}}$

---

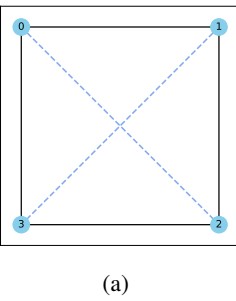 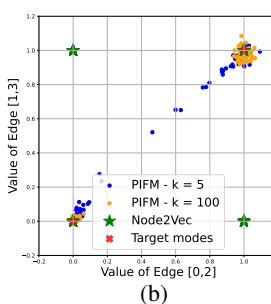 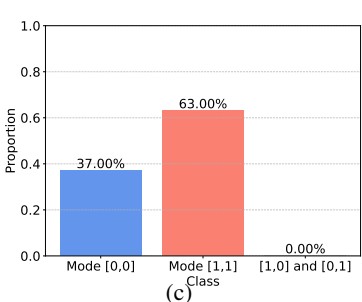

      (a)                    (b)                  (c)

Figure 2: Toy experiment showcasing the advantage of PIFM (in this case, for link prediction). a) Graph $\mathcal{G}$ with four nodes, where the hidden edges are $e_{02}$ and $e_{13}$. b) Generated samples by using node2vec and PIFM (our proposed method): clearly, our method learns a probabilistic coupling, rendering a model that generates only the two valid modes. c) Proportions of samples generated with PIFM from each mode; remarkably, the method also learns a good approximation of the probability of each mode.

0.5, so entries with predicted probability $\geq 0.5$ are treated as edges and those $< 0.5$ as non-edges. In addition, we use maximum mean discrepancy (MMD) (O'Bray et al., 2022) to compute the distance between the generated graphs and the ground truth, serving as a proxy for computing the perception quality. More details on these metrics are deferred to Appendix E.4.

**Baselines.** We compare PIFM against several baselines, including diffusion-based. Recall that PIFM is composed of a one-shot prediction used as prior followed by a flow model. Naturally, we compare PIFM to the accuracy of the one-shot prediction (without the flow) and with a flow with a random starting point:

- **SIGL Prior (Azizpour et al., 2025)/Node2Vec Prior (Grover & Leskovec, 2016)/GraphSAGE Prior (Hamilton et al., 2017)**: one-shot predictions using the structural prior directly.
- **Flow with Gaussian prior**: flow model initialized from uniform Gaussian $\mathcal{N}(0.5, 1)$ noise on masked entries.
- **DiGress + RePaint (Vignac et al., 2023)**: unconditional DiGress combined with RePaint-style resampling (Lugmayr et al., 2022).
- **GDSS + RePaint (Jo et al., 2022)**: unconditional GDSS combined with RePaint-style resampling (Lugmayr et al., 2022).

Algorithmic details of the baselines are provided in Appendix A. Lastly, we consider an additional experiment on a transductive case (CORA Yang et al. (2016)), where we compared with traditional baselines Li et al. (2023).

## 5.2 LINK PREDICTION

Tables 1 and 2 report results for $10\%$ and $50\%$ masking, respectively. Overall, PIFM improves the AUC-ROC of all base priors (SIGL, node2vec, and GraphSAGE). E.g., compare node2vec with PIFM

Table 1: Graph reconstruction performance with **10% of edges masked (0.1 Drop)**. We report AUC, Average Precision (AP↑), False Positive Rate (FPR↓), and False Negative Rate (FNR↓), all in percent (%). The best result for each metric is in **blue** and the second best green.

| | ENZYMES | | | | PROTEINS | | | | IMDB-B | | | |
|---|---|---|---|---|---|---|---|---|---|---|---|---|
| **Method** | AP↑ | AUC↑ | FNR↓ | FPR↓ | AP↑ | AUC↑ | FNR↓ | FPR↓ | AP↑ | AUC↑ | FNR↓ | FPR↓ |
| *Baselines* | | | | | | | | | | | | |
| Node2Vec | 24.62 | 59.60 | 51.39 | 37.96 | 33.24 | 64.40 | 48.56 | 35.37 | 65.00 | 56.36 | 50.27 | 41.68 |
| SIGL | 18.17 | 48.04 | 69.33 | 25.43 | 26.77 | 48.91 | 100.00 | 0.00 | 58.91 | 50.61 | 88.44 | 16.33 |
| GraphSAGE | 41.28 | 73.70 | 13.49 | 60.59 | 46.36 | 74.58 | 11.00 | 63.50 | 83.55 | 83.26 | 16.42 | 36.89 |
| DiGress + RePaint | 33.39 | 67.86 | 58.92 | 5.19 | 40.34 | 72.39 | 47.82 | 6.00 | 59.25 | 58.63 | 76.44 | 7.68 |
| GDSS + RePaint | 18.35 | 47.04 | 74.31 | 32.19 | 26.96 | 51.39 | 63.07 | 32.09 | 57.89 | 46.11 | 69.75 | 36.17 |
| Flow w/ Gaussian prior | 40.09 | 72.44 | 71.03 | 5.87 | 57.86 | 80.83 | 65.09 | 3.07 | 98.89 | 98.37 | 2.26 | 2.54 |
| *Ours* | | | | | | | | | | | | |
| PIFM (Node2Vec) | 41.67 | 76.86 | 72.09 | 5.11 | 58.25 | 81.74 | 59.37 | 6.34 | 97.60 | 97.28 | 1.37 | 3.77 |
| PIFM (GraphSAGE) | 47.21 | 80.25 | 72.85 | 2.40 | 54.79 | 81.02 | 55.73 | 5.40 | 99.37 | 98.79 | 1.81 | 3.37 |
| PIFM (SIGL) | 26.93 | 59.48 | 71.33 | 11.33 | 42.21 | 60.76 | 60.75 | 7.48 | 85.60 | 83.21 | 16.37 | 18.41 |

Table 2: Graph reconstruction performance with **50% of edges masked (0.5 Drop)** (see Table 1 for definitions).

| | ENZYMES | | | | PROTEINS | | | | IMDB-B | | | |
|---|---|---|---|---|---|---|---|---|---|---|---|---|
| **Method** | AP↑ | AUC↑ | FNR↓ | FPR↓ | AP↑ | AUC↑ | FNR↓ | FPR↓ | AP↑ | AUC↑ | FNR↓ | FPR↓ |
| *Baselines* | | | | | | | | | | | | |
| Node2Vec | 19.14 | 55.22 | 46.37 | 44.29 | 23.51 | 53.83 | 51.37 | 44.36 | 54.20 | 52.22 | 48.41 | 47.51 |
| SIGL | 16.88 | 49.30 | 72.01 | 27.85 | 22.90 | 52.55 | 100.00 | 0.00 | 50.05 | 45.41 | 87.68 | 18.80 |
| GraphSAGE | 22.79 | 57.77 | 40.02 | 52.16 | 27.71 | 53.99 | 32.16 | 66.86 | 75.74 | 75.54 | 18.18 | 44.86 |
| DiGress + RePaint | 17.34 | 55.22 | 77.95 | 11.62 | 23.65 | 55.45 | 71.46 | 17.65 | 56.47 | 58.89 | 73.00 | 10.27 |
| GDSS + RePaint | 16.43 | 49.65 | 69.45 | 30.46 | 22.33 | 51.42 | 66.44 | 32.23 | 53.39 | 51.20 | 69.35 | 29.22 |
| Flow w/ Gaussian prior | 17.43 | 51.84 | 98.49 | 1.07 | 26.40 | 55.55 | 93.21 | 5.25 | 78.72 | 79.76 | 41.56 | 14.62 |
| *Ours* | | | | | | | | | | | | |
| PIFM (Node2Vec) | 22.95 | 59.14 | 90.71 | 3.53 | 27.57 | 59.68 | 87.05 | 8.98 | 84.46 | 85.71 | 32.95 | 15.03 |
| PIFM (GraphSAGE) | 25.44 | 61.36 | 95.62 | 1.86 | 35.50 | 60.61 | 85.05 | 10.23 | 93.13 | 93.84 | 17.52 | 7.61 |
| PIFM (SIGL) | 17.08 | 49.15 | 86.06 | 12.28 | 28.38 | 59.58 | 61.20 | 20.38 | 59.83 | 58.11 | 38.90 | 36.76 |

initialized with node2vec. The marked consistent gain can be attributed to the value added by the flow model in capturing the distribution of the true graphs of interest. Moreover, the fact that PIFM with some of the informative priors tends to outperform the flow with a Gaussian prior highlights the value of the two-step procedure advocated here. Among the different priors used, PIFM(GraphSAGE) tends to perform better, especially at a $50\%$ drop rate and in the dense IMDB-B graphs.

For the experiments in this section, the reported PIFM results use $K = 1$, which yields the lowest MSE and, accordingly, the highest AUC-ROC (consistent with the distortion–perception trade-off discussed in Section 4.1). Notably, PIFM with $K = 1$ outperforms the priors (see Appendix F.2 for an ablation of parameters), even though the latter approximate the MMSE estimator, which should be optimal in terms of MSE. While this configuration is optimal for distortion, perceptual quality improves with more steps, as explained below in Section 5.3. Finally, the assumptions in Section 4.2 are quite strong and lead to an approximate MMSE that is not truly optimal, allowing PIFM with $K = 1$ to outperform by capturing global information that the different priors miss.

## 5.3 BLIND GRAPH RECONSTRUCTION

We focus on two blind versions of link prediction, namely expansion and denoising. In the expansion case, we only get to observe a subset of the edges (but no non-edges), and we need to determine which other entries correspond to existing edges. Conversely, for denoising, we get to observe a subset of the non-edges (but no actual edge), and we need to determine which other entries correspond to non-edges. These cases are more challenging than link prediction since transductive priors like node2vec cannot be trained on the masked graphs (since we do not have positive and negative edges). We present here the results for expansion. The results for denoising can be found in Appendix F.1.

**Expansion.** The goal in expansion is to predict a set of hidden edges $\mathcal{E}_M$ given $\mathbf{A}^{\mathcal{O}}$, such that the edge set of the ground truth is $\mathcal{E} = \mathcal{E}_M \cup \mathcal{E}_O$. Therefore, defining $\mathbf{A}_1 = \mathbf{A}$, the initialization becomes $\mathbf{A}_0 = \mathbf{A}_1^{\mathcal{O}} + (1 - \mathbf{A}_1^{\mathcal{O}}) \odot (f_{prior}(\mathbf{A}_1^{\mathcal{O}}) + \boldsymbol{\epsilon}_s)$. The results for a drop rate of 50% are shown in Table 3. Among all baselines, PIFM (GraphSAGE) attains the top AUC/AP on most of the metrics, surpassing both the GraphSAGE prior and other diffusion baselines. Compared to a Gaussian start, the informed

Table 3: Performance for the **expansion** task with **50% of edges masked (0.5 Drop)** (see Table 1 for definitions).

| | ENZYMES | | | | PROTEINS | | | | IMDB-B | | | |
|---|---|---|---|---|---|---|---|---|---|---|---|---|
| Method | AP↑ | AUC↑ | FNR↓ | FPR↓ | AP↑ | AUC↑ | FNR↓ | FPR↓ | AP↑ | AUC↑ | FNR↓ | FPR↓ |
| *Baselines* | | | | | | | | | | | | |
| GraphSAGE | **13.95** | 57.54 | **40.29** | 52.74 | 18.91 | 53.91 | **31.62** | 67.18 | 67.18 | 74.92 | **19.04** | 45.20 |
| DiGress + RePaint | 2.41 | 54.25 | 84.03 | 7.54 | 4.52 | 56.87 | 81.10 | 13.00 | 22.93 | 56.37 | 81.05 | **6.69** |
| GDSS + RePaint | 9.21 | 49.63 | 69.45 | 30.80 | 14.66 | 51.03 | 66.44 | 32.06 | 39.43 | 50.68 | 69.35 | 30.00 |
| Flow w/ Gaussian prior | 9.45 | 50.40 | 90.64 | 9.35 | 14.71 | 50.31 | 82.32 | 17.28 | 49.46 | 62.28 | 71.41 | 13.64 |
| *Ours* | | | | | | | | | | | | |
| PIFM (GraphSAGE) | 13.17 | **60.09** | 100.00 | **0.00** | **21.70** | **62.34** | 94.75 | **4.54** | **83.49** | **87.28** | 29.74 | 11.27 |

prior is crucial to improve AUC and AP, indicating effective global coupling beyond local scores. Overall, PIFM serves as a better reconstructor in this challenging case, with $K$ providing a tunable perception–distortion trade-off (cf. Appendix F.3).

**Distortion-perception trade-off.** While a single-step reconstruction ($K = 1$) yields the lowest distortion (AUC-ROC), we assess if more steps improve perceptual quality. We measure the $MMD^2$ score between the generated and ground-truth graph distributions on the ENZYMES dataset as a function of the number of steps, $K$. As shown in Fig. 3(a), the $MMD^2$ score decreases as $K$ increases, signifying a closer match to the true data distribution and thus higher realism. We further validate this by comparing graph statistics (degree, triangles, clustering coefficients), which also show that a larger $K$ more closely matches the ground-truth. Additional results and details are in Appendices F.3 and F.4.

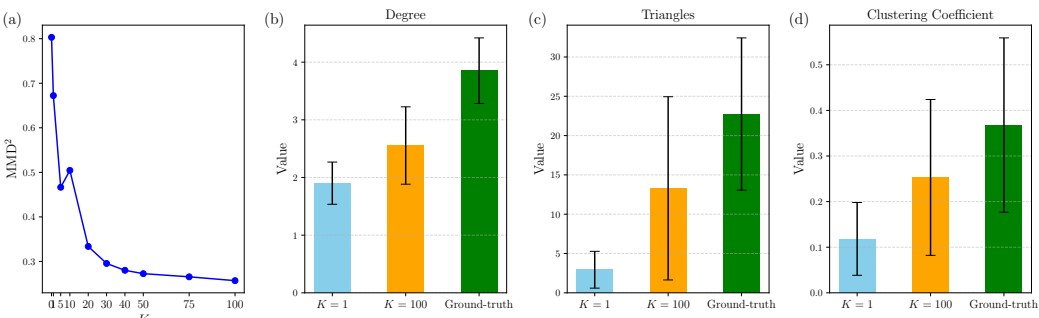

Figure 3: Increasing the number of reconstruction steps ($K$) improves perceptual quality. (a) The MMD score, measuring the distance to the true data distribution, decreases as $K$ increases. (b-d) This result is corroborated by key graph statistics, where the average degree, number of triangles, and clustering coefficient for graphs generated with $K = 100$ more closely match the ground-truth distribution compared to those generated with $K = 1$. Error bars indicate the standard deviation over 300 samples (10 samples for each of the 30 test graphs).

## 6 CONCLUSIONS

In this paper, we introduced Prior-Informed Flow Matching (PIFM), a method for graph reconstruction that learns global structural information by integrating local edge predictors within a flow-based generative model. PIFM formulates graph topology inference as a distortion-perception problem, learning an optimal transport map from a local estimator to the ground-truth graph distribution. We evaluate PIFM using two types of local estimators, inductive (graphons and graphSAGE) and transductive (node2vec), which induce different reconstruction behaviors. Experiments on multiple benchmark datasets show that PIFM consistently outperforms both classical embedding methods and recent flow-based baselines, demonstrating the significant value of learning global edge correlations.

Our method has limitations, primarily inheriting the scalability challenges of diffusion models in graphs. Future work could explore sub-graph-based alternatives to improve efficiency (Trivedi et al., 2024). Additionally, our current formulation is limited to homogeneous graphs; extending PIFM to heterogeneous graphs by defining the process in the probability simplex (Eijkelboom et al., 2024) or using discrete flow models Qin et al. (2025) is another promising direction for future research.

## REPRODUCIBILITY STATEMENT

The experimental setups and results are detailed in Section 5 of the main paper. Further specifics, including comprehensive dataset descriptions, additional experimental details, and ablation studies, are provided in Appendix E. Furthermore, and to facilitate full reproducibility, we include a complete codebase as supplementary material. This supplementary package contains clearly organized configuration files (e.g., YAML files) that detail all hyperparameters used across our experiments, enabling straightforward replication of our reported findings.

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

# A  ALGORITHM

In this section, we describe the algorithms that we use as baselines. Each method serves distinct purposes: SIGL/node2vec/GraphSAGE $A_0$ tests whether the flow model provides meaningful improvement beyond the one-shot estimates given by the priors, uniform + flow evaluates whether the SIGL/node2vec/GraphSAGE predicted graphons are good structural priors for effective denoising, and DiGress + RePaint compares our model to standard modified unconditionally generation models.

## A.1  UNIFORM + FLOW BASELINE

This baseline ablates the structural prior by initializing the flow from a state where unknown entries are filled with uniform noise. The model then learns to denoise from this less-informed starting point.

---
**Algorithm 2** Uniform + flow Training and Sampling
---

**Training**
1: Sample $\mathbf{A}_1$, a mask $\xi$, and time $t \sim U[0, 1]$.
2: Define initial state with Gaussian noise added to the masked region:

$$\mathbf{A}_0 \triangleq \xi \odot \mathbf{A}_1 + (1 - \xi) \odot \mathcal{U}(0, 1)^{N \times N} + (1 - \xi) \odot \boldsymbol{\epsilon}_{\text{train}}, \quad \boldsymbol{\epsilon}_{\text{train}} \sim \mathcal{N}(0, \sigma_{\text{train}}^2).$$

3: Define interpolant $\mathbf{A}_t \triangleq (1 - t)\mathbf{A}_0 + t\mathbf{A}_1$.
4: Solve $\theta^* = \operatorname{argmin}_\theta \mathbb{E}_{\mathbf{A}_1, \xi, t} \| v_\theta(\mathbf{A}_t, t) - (\mathbf{A}_1 - \mathbf{A}_0) \|_F^2$.
   **Sampling (Reconstruction)**
5: Given observed graph $\mathbf{A}_1^{\mathcal{O}}$, define the initial state with masked noise:

$$\hat{\mathbf{A}} \leftarrow \xi \odot \mathbf{A}_1^{\mathcal{O}} + (1 - \xi) \odot \mathcal{U}(0, 1)^{N \times N} + (1 - \xi) \odot \boldsymbol{\epsilon}_{\text{samp}}, \quad \boldsymbol{\epsilon}_{\text{samp}} \sim \mathcal{N}(0, \sigma_{\text{samp}}^2).$$

6: **for** $i \leftarrow 0, \ldots, K - 1$ **do**
7:     $\hat{\mathbf{A}} \leftarrow \hat{\mathbf{A}} + \frac{1}{K} v_{\theta^*}\left(\hat{\mathbf{A}}, \frac{i}{K}\right)$
8: Return $\hat{\mathbf{A}}$

---

## A.2  DIGRESS + REPAINT BASELINE

**Training (Unconditional)**  The model $p_\theta$ is trained unconditionally on complete graphs $\mathbf{A}_1 \sim p_{\text{data}}$ to reverse a discrete forward noising process $q$. The forward process is a fixed Markov chain $q(\mathbf{A}_t | \mathbf{A}_{t-1})$ that corrupts the graph over $T$ steps. The training objective is to learn the denoising distribution $p_\theta(\mathbf{A}_1 | \mathbf{A}_t)$, modeled as a categorical prediction task for each node and edge.

---
**Algorithm 3** DiGress Unconditional Training
---

**Forward Process:** Sample a noised graph at any timestep $t$ directly via $\mathbf{A}_t \sim q(\mathbf{A}_t | \mathbf{A}_1)$.
**Denoising Objective:**
1: Train a denoising network $p_\theta(\cdot, t)$ to predict the original graph $\mathbf{A}_1$ from $\mathbf{A}_t$.
2: Minimize the expected cross-entropy loss w.r.t. the ground truth:

$$\theta^* = \operatorname*{argmin}_{\theta} \mathbb{E}_{\mathbf{A}_1 \sim p_{\text{data}}, t \sim \mathcal{U}\{1..T\}} \left[\mathcal{L}_{\text{CE}}\left(\mathbf{A}_1, p_\theta(\mathbf{A}_t, t)\right)\right]$$

---

**Sampling (Conditional Reconstruction via RePaint)**  At inference, given an observed graph $\mathbf{A}_1^{\mathcal{O}} \triangleq \xi \odot \mathbf{A}_1$, the unconditionally trained model $p_{\theta^*}$ generates the missing entries. This is achieved by iteratively re-imposing the known (unmasked) information during the reverse diffusion process (Lugmayr et al., 2022).

---

**Algorithm 4** DiGress + RePaint Sampling

---

    **Input:** Observed graph $\mathbf{A}_1^{\mathcal{O}}$, mask $\xi$, trained model $p_{\theta*}$, steps $T$.
    **Output:** Reconstructed graph $\hat{\mathbf{A}}_1$.
  1: Initialize $\hat{\mathbf{A}}_T \sim p_{\text{prior}}(\cdot)$, where $p_{\text{prior}}$ is a random graph distribution.
  2: **for** $t = T, T-1, \ldots, 1$ **do**
  3:     *// Predict clean graph from current state*
  4:     $\tilde{\mathbf{A}}_1 = p_{\theta*}(\hat{\mathbf{A}}_t, t)$.
  5:
  6:     *// Impose known data by noising it to the current step*
  7:     $\mathbf{A}_t^{\text{known}} \sim q(\mathbf{A}_t | \mathbf{A}_1^{\mathcal{O}})$.
  8:
  9:     *// Sample the unknown region by noising the prediction to the next step*
10:     $\mathbf{A}_{t-1}^{\text{unknown}} \sim q(\mathbf{A}_{t-1} | \tilde{\mathbf{A}}_1)$.
11:
12:     *// Combine known and unknown parts for the next state*
13:     $\hat{\mathbf{A}}_{t-1} = \xi \odot \mathbf{A}_t^{\text{known}} + (1 - \xi) \odot \mathbf{A}_{t-1}^{\text{unknown}}$.
14: **end for**
15: **return** $p_{\theta*}(\hat{\mathbf{A}}_1, 1)$

---

### A.3 NODE2VEC PRIOR (PER-GRAPH CLASSIFIER)

This baseline learns a *per-graph* edge-probability model from the observed subgraph. We (i) fit node2vec embeddings on the observed topology and (ii) train a logistic classifier on Hadamard edge features to produce probabilities on the masked pairs.

---

**Algorithm 5** Node2Vec Prior: Training and Inference

---

    **Inputs:** Full adjacency $\mathbf{A}_1$, mask $\xi$ ($\xi_{ij} = 1$ if observed), Node2Vec hyperparams (dim $d$, walk length $L$, walks/node $R$, window $w$, $p, q$), negatives/positive ratio $k$.
    **Outputs:** Probabilities $\hat{P}$ on masked entries, i.e., $f_{\text{prior}}(\mathbf{A}_1^{\mathcal{O}})$.
    **Training (per graph)**
  1: Construct observed graph $\mathbf{A}_1^{\mathcal{O}} \leftarrow \xi \odot \mathbf{A}_1$.
  2: Train Node2Vec on $\mathbf{A}_1^{\mathcal{O}}$ to obtain node embeddings $\{\mathbf{z}_i\}_{i=1}^N \in \mathbb{R}^d$.
  3: Build labeled edge set on *observed* pairs (upper triangle $i < j$):

$$\mathcal{P}^+ = \{(i,j) : \xi_{ij} = 1, A_{ij} = 1\}, \quad \mathcal{P}^- \sim k\text{-to-1 balanced samples from } \{(i,j) : \xi_{ij} = 1, A_{ij} = 0\}.$$

  4: Features: $\mathbf{x}_{ij} \leftarrow \mathbf{z}_i \odot \mathbf{z}_j$ (Hadamard product);     Labels: $y_{ij} \in \{0, 1\}$.
  5: Fit a logistic classifier $g_\phi(\mathbf{x}) = \sigma(\mathbf{w}^\top \mathbf{x} + b)$ (L2-regularized; class-balanced).
    **Inference (per graph)**
  6: For each *masked* pair $(i, j)$ with $\xi_{ij} = 0$, compute $\mathbf{x}_{ij} \leftarrow \mathbf{z}_i \odot \mathbf{z}_j$.
  7: Predict $\hat{P}_{ij} \leftarrow g_\phi(\mathbf{x}_{ij})$ and set $\hat{P}_{ji} \leftarrow \hat{P}_{ij}$.
  8: Return $\hat{P}$ as $f_{\text{prior}}(\mathbf{A}_1^{\mathcal{O}})$ (used in Eq. (8)).

---

*Notes.* (i) We train embeddings *only* on $\mathbf{A}_1^{\mathcal{O}}$ to avoid leakage. (ii) The Hadamard feature works well and is symmetric; concatenation can be used but breaks symmetry unless sorted. (iii) Thresholding at 0.5 yields hard reconstructions; we use scores $\hat{P}$ directly in PIFM.

## B BACKGROUND

### B.1 GRAPHONS AND GRAPHON ESTIMATION

As described in Section 4.2, a graphon is defined as a bounded, symmetric, and measurable function $\mathcal{W} : [0, 1]^2 \rightarrow [0, 1]$ (Lovász, 2012). By construction, a graphon acts as a *generative model for random graphs*, allowing the sampling of graphs that exhibit similar structural properties. To generate an undirected graph $\mathcal{G}$ with $N$ nodes from a given graphon $\mathcal{W}$, the process consists of two main steps:

(1) assigning each node a latent variable drawn uniformly at random from the interval $[0, 1]$, and (2) connecting each pair of nodes with a probability given by evaluating $\mathcal{W}$ at their respective latent variable values.

The generative process in (7) can also be viewed in reverse: given a collection of graphs (represented by their adjacency matrix) $\mathcal{D} = \{\mathbf{A}_t\}_{t=1}^{M}$ that are sampled from an *unknown* graphon $\mathcal{W}$, estimate $\mathcal{W}$. Several methods have been proposed for this task (Chan & Airoldi, 2014; Airoldi et al., 2013; Xu et al., 2021; Xia et al., 2023; Azizpour et al., 2025). We focus on SIGL (Azizpour et al., 2025), a resolution-free method that, in addition to estimating the graphon, also *infers the latent variables* $\boldsymbol{\eta}$, making it particularly useful for model-driven augmentation in GCL. This method parameterizes the graphon using an implicit neural representation (INR) (Sitzmann et al., 2020), a neural architecture defined as $f_\phi(x, y) : [0, 1]^2 \rightarrow [0, 1]$ where the inputs are coordinates from $[0, 1]^2$ and the output approximates the graphon value $\mathcal{W}$ at a particular position. In a nutshell, SIGL works in three steps: (1) a sorting step using a GNN $g_{\phi'}(\mathbf{A})$ that estimates the latent node positions or representations $\boldsymbol{\eta}$; (2) a histogram approximation of the sorted adjacency matrices; and (3) learning the parameters $\phi$ by minimizing the mean squared error between $f_\phi(x, y)$ and the histograms (obtained in step 2). More details of SIGL are provided in Appendix B.1.

## B.2 NODE2VEC

*node2vec* (Grover & Leskovec, 2016) is a scalable model for learning continuous node representations in graphs. This methods is *transductive*, meaning that it generates an embedding per graph. It extends the Skip-gram model from natural language processing to networks by sampling sequences of nodes through biased random walks. Node2vec introduces two hyperparameters $(p, q)$ that interpolate between breadth-first and depth-first exploration. This flexibility allows embeddings to capture both *homophily* (nodes in the same community) and *structural equivalence* (nodes with similar roles, e.g., hubs), which frequently coexist in real-world graphs.

The embeddings are learned via stochastic gradient descent with negative sampling to maximize the likelihood of preserving sampled neighborhoods. Once learned, node embeddings can be combined through simple binary operators (e.g., Hadamard product) to form edge features, enabling applications such as link prediction. Empirically, node2vec has been shown to outperform prior unsupervised embedding methods across tasks like classification and link recovery, while remaining computationally efficient and scalable to large graphs (Grover & Leskovec, 2016).

## B.3 GRAPHSAGE

GraphSAGE (Hamilton et al., 2017) is an *inductive* technique for link prediction based on graph neural networks (GNN) framework designed to generate embeddings for nodes in large, evolving graphs. It consists of two-steps: for a target node, it first samples a fixed-size neighborhood of adjacent nodes, and then it aggregates feature information from these sampled neighbors. By learning aggregation functions (such as a mean, pool, or LSTM aggregator) rather than embeddings for every single node, GraphSAGE can efficiently generate predictions for nodes that were not part of the training set, making it highly scalable and effective for real-world applications like social networks and recommendation systems.

## B.4 GRAPH DIFFUSION MODELS

Diffusion models are generative frameworks composed by two processes: a **forward process** that systematically adds noise to data until it becomes pure noise, and a **reverse process** that learns to reverse this, generating new data by starting from noise and progressively denoising it. While these models exist for both discrete (Vignac et al., 2023) and continuous domains (Jo et al., 2022), we describe the continuous case which is the most related to our method. Here, a graph $\mathbf{G}_0$ is defined by its node features $\mathbf{X}_0 \in \mathbb{R}^{N \times F}$ and its weighted adjacency matrix $\mathbf{A}_0 \in \mathbb{R}^{N \times N}$. Following the GDSS framework, the forward process is described by a stochastic differential equation (SDE) that gradually perturbs the graph data over a time interval $t \in [0, T]$:

$$\mathrm{d}\mathbf{G}_t = -\frac{1}{2}\beta(t)\mathbf{G}_t \, \mathrm{d}t + \sqrt{\beta(t)} \, \mathrm{d}\mathbf{W}_t$$

In this equation, $\mathbf{W}_t$ represents standard Brownian motion (i.e., noise), and $\beta(t)$ is a noise schedule that typically increases over time. This process is designed so that by the final time $T$, the original data distribution $\mathbf{G}_T$ is indistinguishable from a standard Gaussian.

The generative reverse process is defined by another SDE that traces the path from noise back to data. This process relies on the **score function**, $\nabla_{\mathbf{G}_t} \log p(\mathbf{G}_t)$, which is the gradient of the log-density of the noisy data at time $t$. Since the true score function is unknown, it must be approximated. This is done using a neural network, or **score network**, which is trained to predict the score. For graphs, separate networks are often used for the adjacency matrix and node features: $\epsilon_{\boldsymbol{\theta}_A}(\mathbf{A}_t, t)$ and $\epsilon_{\boldsymbol{\theta}_X}(\mathbf{X}_t, t)$. These networks are trained by minimizing the denoising score-matching loss.

Once trained, these score networks can be plugged into the reverse SDE. New graphs are then generated by solving this SDE numerically using standard samplers like DDPM or DDIM.

## C  RELATED WORKS

### C.1  LINK PREDICTION

Link prediction aims to determine if an unobserved edge should exist between two nodes within a partially observed graph (Newman, 2001; Adamic & Adar, 2003; Zhou et al., 2009). Classical approaches rely on topology-only heuristics.

More recently, unsupervised node embedding methods have become an effective strategy for link prediction. These methods learn a low-dimensional vector for each node that represents neighborhood similarity and community structure, often using random walks and an objective similar to Skip-gram. Consequently, nodes that are close in the embedding space are more likely to be linked. DeepWalk was a pioneering method that modeled short random walks to learn generalizable representations for tasks like predicting missing links (Perozzi et al., 2014). Node2vec builds on DeepWalk by employing biased, second-order random walks to balance breadth-first and depth-first searches and by converting node embeddings into edge features. In node2vec, embeddings for nodes $f(u)$ and $f(v)$ are combined with binary operators to create an edge representation $g(u, v)$, which a classifier then uses to determine if the edge $(u, v)$ exists (Grover & Leskovec, 2016).

Graph neural networks (GNNs) are also widely used for edge reconstruction. A typical encoder-decoder framework uses message passing to learn node embeddings and a simple decoder to generate link scores. Inductive frameworks like GraphSAGE learn functions to sample and aggregate features from a node's neighborhood, allowing the model to generalize to new nodes or graphs (Hamilton et al., 2017). A different approach focuses on modeling the pair representation directly. For example, Neural Bellman-Ford Networks (NBFNet) frame link prediction as a path-aggregation problem. The score for a pair of nodes is calculated as the sum of all path representations between them, with each path being a product of its edge representations. This formulation is solved using a generalized Bellman-Ford iteration, where NBFNet parameterizes the operators with neural functions, creating an interpretable and inductive framework (Zhu et al., 2021).

### C.2  DIFFUSION-BASED INVERSE PROBLEMS SOLVER FOR GRAPHS

We now expand on diffusion-based solvers for graph inverse problems. Given a condition $\mathcal{C}$ and a reward function $r(\mathbf{G}_0)$ that quantifies how close the sample $\mathbf{G}_0$ is to meeting $\mathcal{C}$, the objective is to generate graphs $G_0$ that maximize the reward function. From a Bayesian perspective, this problem boils down to sampling from the posterior $p(\mathbf{G}_0|\mathcal{C}) \propto p(\mathcal{C}|\mathbf{G}_0)p(\mathbf{G}_0)$ where $p(\mathcal{C}|\mathbf{G}_0) \propto \exp(r(\mathbf{G}_0))$ is a likelihood term and $p(\mathbf{G}_0)$ is a prior given by the pre-trained diffusion model. We now describe previous works for both differentiable and non-differentiable reward functions.

**Guidance with Differentiable Reward Functions.**  Several approaches have been developed to guide generative models when the objective can be expressed as a **differentiable reward function**, particularly for inverse problems in imaging. These methods typically leverage the differentiability of the reward – often a likelihood tied to a noisy measurement – to calculate a *conditional score* using Bayes' rule:

$$\nabla_{\mathbf{G}_t} \log p(\mathbf{G}_t|\mathcal{C}) = \nabla_{\mathbf{G}_t} \log p(\mathcal{C}|\mathbf{G}_t) + \nabla_{\mathbf{G}_t} \log p(\mathbf{G}_t)$$

In this formulation, the diffusion model naturally serves as the prior ($p(\mathbf{G}_t)$), while the likelihood term ($p(\mathcal{C}|\mathbf{G}_t)$) provides the guidance. However, directly computing the score of the likelihood term is intractable because it requires integrating over all possible clean data: $p(\mathcal{C}|\mathbf{G}_t) = \int p(\mathcal{C}|\mathbf{G}_0)p(\mathbf{G}_0|\mathbf{G}_t)d\mathbf{G}_0$.

To overcome this, a common technique is to approximate the posterior distribution $p(\mathbf{G}_0|\mathbf{G}_t)$ with a Gaussian centered at the MMSE denoiser. This denoised estimate can be calculated efficiently using **Tweedie's formula**:

$$\mathbb{E}[\mathbf{G}_0|\mathbf{G}_t] = \frac{1}{\alpha_t}\left(\mathbf{G}_t + \sigma_t^2 \nabla_{\mathbf{G}_t} \log p(\mathbf{G}_t, t)\right)$$

While this framework is established for images, its application to graph-based inverse problems is less explored. This is primarily because most interesting properties and constraints in graphs are **not differentiable**. Some graph-specific methods, like DiGress (Vignac et al., 2023), implement guidance by training an auxiliary model, similar to classifier-free guidance, which introduces additional complexity.

**Guidance with Non-Differentiable Reward Functions.** For the more common scenario of non-differentiable constraints in graph generation, alternative strategies have emerged. The **PRODIGY** method, for instance, operates by repeatedly applying a two-step process at each denoising step: generation followed by projection.

First, it uses the unconditional diffusion model to produce a candidate sample $\hat{\mathbf{G}}_{t-1}$. Second, it projects this candidate onto the set of valid solutions using a projection operator: $\Pi_{\mathcal{C}}(\hat{\mathbf{G}}_{t-1}) = \arg\min_{\mathbf{Z}\in\mathcal{C}}\|\mathbf{Z} - \hat{\mathbf{G}}_{t-1}\|_2^2$. Since applying the full projection at every step can destabilize the generation process, PRODIGY uses a partial update to balance constraint satisfaction with the learned diffusion trajectory:

$$\mathbf{G}_{t-1} \leftarrow (1 - \gamma_t)\hat{\mathbf{G}}_{t-1} + \gamma_t \Pi_{\mathcal{C}}(\hat{\mathbf{G}}_{t-1})$$

This approach has two main limitations. First, it is only practical for simple constraints where the projection operator $\Pi_{\mathcal{C}}(\cdot)$ has an efficient, closed-form solution. Second, it applies the projection directly to the noisy intermediate sample $\mathbf{G}_t$, whereas the constraint $\mathcal{C}$ is defined on the clean data $\mathbf{G}_0$, creating a domain mismatch. Recently, in Tenorio et al. (2025), the authors leverage zeroth-order optimizaton to build a guidance term, improving over PRODIGY in challenging tasks.

### C.3 FLOW-BASED INVERSE SOLVERS

More recently, two flow-based generative models for graphs have been proposed. Catflow, introduced in Eijkelboom et al. (2024), formulates flow matching as a variational inference problem, allowing to build a model for categorical data. The key difference between Catflow and traditional flow matching is that in the former, the objective is to approximate the posterior probability path, which is a distribution over possible end points of a trajectory. Compared to discrete diffusion, this formulation defines a path in the probability simplex, building a continuous path. This formulation boils down to a cross-entropy loss. Another recent work is DeFoG, introduced in Qin et al. (2025). This method is inspired by discrete flow matching (Campbell et al., 2024), where a discrete probability path is used. Similarly, the loss is a cross-entropy.

## D PROOFS

### D.1 PROOF FOR THEOREM 1

*Proof.* Our goal is to show that for any permutation matrix $\mathbf{P}$, our estimated density satisfies $\log p(\mathbf{P}^\top \mathbf{A}_1 \mathbf{P}) = \log p(\mathbf{A}_1)$. First, we notice that $\text{tr}\left(\frac{\partial v_\theta(\mathbf{A}_t, t)}{\partial \mathbf{A}_t}\right) = \langle v_\theta(\mathbf{A}_t, t), d\mathbf{A}_t\rangle_F$ Let's define a permuted graph $\mathbf{A}_1' = \mathbf{P}^\top \mathbf{A}_1 \mathbf{P}$ and a similarly permuted mask $\xi' = \mathbf{P}^\top \xi \mathbf{P}$; to simplify notation, we denote $F(.) = \log p(.)$.

First, we establish the equivariance of the initial state $\mathbf{A}_0$. Let $\mathbf{A}'_0$ be the initial state constructed from the permuted graph $\mathbf{A}'_1$ and mask $\xi'$.

$$
\begin{aligned}
\mathbf{A}'_0 &= \xi' \odot \mathbf{A}'_1 + (1 - \xi') \odot f_{\text{prior}}(\xi' \odot \mathbf{A}'_1) \\
&= (\mathbf{P}^\top \xi \mathbf{P}) \odot (\mathbf{P}^\top \mathbf{A}_1 \mathbf{P}) + (1 - \mathbf{P}^\top \xi \mathbf{P}) \odot f_{\text{prior}}((\mathbf{P}^\top \xi \mathbf{P}) \odot (\mathbf{P}^\top \mathbf{A}_1 \mathbf{P})) \\
&= \mathbf{P}^\top (\xi \odot \mathbf{A}_1) \mathbf{P} + \mathbf{P}^\top (1 - \xi) \mathbf{P} \odot f_{\text{prior}}(\mathbf{P}^\top (\xi \odot \mathbf{A}_1) \mathbf{P}) \quad (\text{since } \odot \text{ distributes over } \mathbf{P})
\end{aligned}
$$

The key requirement for this proof is the permutation equivariance of the prior estimator, $f_{\text{prior}}$. This condition is satisfied by both prior models used in our work. Our SIGL-based prior is permutation equivariant by design, as it uses a GNN encoder to learn the graphon structure. Our node2vec-based prior enforces permutation equivariance by first mapping nodes to a canonical ordering based on the principal components of their embeddings, ensuring that any permutation of an input graph is processed identically.

With the permutation equivariance of $f_{\text{prior}}$ established, such that $f_{\text{prior}}(\mathbf{P}^\top \mathbf{X} \mathbf{P}) = \mathbf{P}^\top f_{\text{prior}}(\mathbf{X}) \mathbf{P}$, we can apply this property:

$$
\begin{aligned}
\mathbf{A}'_0 &= \mathbf{P}^\top (\xi \odot \mathbf{A}_1) \mathbf{P} + \mathbf{P}^\top (1 - \xi) \mathbf{P} \odot (\mathbf{P}^\top f_{\text{prior}}(\xi \odot \mathbf{A}_1) \mathbf{P}) \\
&= \mathbf{P}^\top (\xi \odot \mathbf{A}_1) \mathbf{P} + \mathbf{P}^\top ((1 - \xi) \odot f_{\text{prior}}(\xi \odot \mathbf{A}_1)) \mathbf{P} \\
&= \mathbf{P}^\top (\xi \odot \mathbf{A}_1 + (1 - \xi) \odot f_{\text{prior}}(\xi \odot \mathbf{A}_1)) \mathbf{P} \\
&= \mathbf{P}^\top \mathbf{A}_0 \mathbf{P}
\end{aligned}
$$

Thus, the initial state $\mathbf{A}_0$ is permutation-equivariant.

Next, we examine the flow path $\mathbf{A}'_t$ corresponding to the permuted graph $\mathbf{A}'_1$:

$$
\begin{aligned}
\mathbf{A}'_t &= (1 - t)\mathbf{A}'_0 + t\mathbf{A}'_1 \\
&= (1 - t)(\mathbf{P}^\top \mathbf{A}_0 \mathbf{P}) + t(\mathbf{P}^\top \mathbf{A}_1 \mathbf{P}) \\
&= \mathbf{P}^\top ((1 - t)\mathbf{A}_0 + t\mathbf{A}_1) \mathbf{P} \\
&= \mathbf{P}^\top \mathbf{A}_t \mathbf{P}.
\end{aligned}
$$

The path itself is equivariant. The differential element also transforms equivariantly: $d\mathbf{A}'_t = \mathbf{P}^\top d\mathbf{A}_t \mathbf{P}$.

Now, we evaluate the scalar function $F(\mathbf{A}'_1)$ by integrating along the permuted path $\mathbf{A}'_t$:

$$
F(\mathbf{A}'_1) = -\int_0^1 \langle v_\theta(\mathbf{A}'_t, t), d\mathbf{A}'_t \rangle_F \, dt + C
$$

Substituting the equivariant forms for the path and its differential:

$$
F(\mathbf{A}'_1) = -\int_0^1 \langle v_\theta(\mathbf{P}^\top \mathbf{A}_t \mathbf{P}, t), \mathbf{P}^\top d\mathbf{A}_t \mathbf{P} \rangle_F \, dt + C
$$

By the assumed permutation equivariance of the velocity field $v_\theta$, we have $v_\theta(\mathbf{P}^\top \mathbf{A}_t \mathbf{P}, t) = \mathbf{P}^\top v_\theta(\mathbf{A}_t, t) \mathbf{P}$. Substituting this in:

$$
F(\mathbf{A}'_1) = -\int_0^1 \langle \mathbf{P}^\top v_\theta(\mathbf{A}_t, t) \mathbf{P}, \mathbf{P}^\top d\mathbf{A}_t \mathbf{P} \rangle_F \, dt + C
$$

The Frobenius inner product $\langle \mathbf{A}, \mathbf{B} \rangle_F = \text{tr}(\mathbf{A}^\top \mathbf{B})$ is invariant to unitary transformations. Specifically, for any orthogonal matrix $\mathbf{P}$ (where $\mathbf{P}^\top \mathbf{P} = \mathbf{I}$):

$$
\begin{aligned}
\langle \mathbf{P}^\top \mathbf{X} \mathbf{P}, \mathbf{P}^\top \mathbf{Y} \mathbf{P} \rangle_F &= \text{tr}((\mathbf{P}^\top \mathbf{X} \mathbf{P})^\top (\mathbf{P}^\top \mathbf{Y} \mathbf{P})) \\
&= \text{tr}(\mathbf{P}^\top \mathbf{X}^\top \mathbf{P} \mathbf{P}^\top \mathbf{Y} \mathbf{P}) \\
&= \text{tr}(\mathbf{P}^\top \mathbf{X}^\top \mathbf{Y} \mathbf{P}) \\
&= \text{tr}(\mathbf{X}^\top \mathbf{Y}) \\
&= \langle \mathbf{X}, \mathbf{Y} \rangle_F.
\end{aligned}
$$

Applying this property, the integrand simplifies:

$$\langle \mathbf{P}^\top v_\theta(\mathbf{A}_t, t)\mathbf{P}, \mathbf{P}^\top d\mathbf{A}_t\mathbf{P}\rangle_F = \langle v_\theta(\mathbf{A}_t, t), d\mathbf{A}_t\rangle_F$$

The integrand is identical for both the original and permuted inputs. Therefore, the integrals are equal:

$$F(\mathbf{P}^\top \mathbf{A}_1 \mathbf{P}) = -\int_0^1 \langle v_\theta(\mathbf{A}_t, t), d\mathbf{A}_t\rangle_F dt + C = F(\mathbf{A}_1)$$

This confirms that the scalar function $F$ is permutation-invariant. □

## E EXPERIMENTAL DETAILS

### E.1 DETAILS ABOUT THE ARCHITECTURE

Our model adopts a modified version of the adjacency score network architecture introduced in GDSS (Jo et al., 2022). The network is a permutation-equivariant graph neural network designed to approximate the scores $\nabla_{\mathbf{A}_t} \log p_t(X_t, \mathbf{A}_t)$ and $\nabla_{\mathbf{x}_t} \log p_t(\mathbf{x}_t, \mathbf{A}_t)$ at each diffusion step; in this paper, we use only score w.r.t. $\mathbf{A}_t$. Concretely, the architecture consists of stacked message-passing layers followed by a multi-layer perceptron. Each layer propagates node and edge information through adjacency-based aggregation, ensuring equivariance under node relabeling. Time information $t$ is incorporated by scaling intermediate activations with the variance of the forward diffusion process, following the practice in continuous-time score models. Residual connections and normalization layers are used to stabilize training. The final output is an $N \times N$ tensor matching the adjacency dimension. This design provides the required permutation-equivariance and expressive power while remaining computationally tractable for mid-sized benchmark graphs.

The modification that incorporates is a module to build an embedding for the variable $t$ and a FiLM style modulation to incorporate noise conditioning. In particular, we incorporate the following modules:

- A positional encoding based on a sinusoidal embedding following Karras et al. (2022)

- An MLP layer with SiLU activation per attention layer

- A modulation at each attention layer, where we scale the hidden features by an adaptive RMS norm operation (Crowson et al., 2024)

### E.2 DETAILS ABOUT THE DATASETS

In Table 4 we report the statistics of the datasets used in the main text.

Table 4: Statistics of the datasets used for evaluation.

| Dataset | # Graphs | Avg. Nodes | Avg. Edges | # Classes | Domain |
|---------|----------|-----------|-----------|-----------|--------|
| ENZYMES | 600 | 32.63 | 62.14 | 6 | Bioinformatics |
| PROTEINS | 1,113 | 39.06 | 72.82 | 2 | Bioinformatics |
| IMDB-B | 1,000 | 19.77 | 96.53 | 2 | Social Network |

### E.3 HYPERPARAMETERS

#### E.3.1 FLOW-BASED BASELINES

We report the hyperparameters governing the model and training. All three baselines use the same rectified-flow architecture and optimizer family; the only substantive differences are the prior settings.

| Node2Vec prior (per-graph link predictor) | |
|---|---|
| `n2v_dim` | 64 |
| `n2v_walk_length` | 30 |
| `n2v_walks_per_node` | 10 |
| `n2v_context` | 10 |
| `n2v_p`, `n2v_q` | 1.0, 1.0 |
| `n2v_epochs` | 1000 |
| `clf_epochs` | 1000 |

| PIFM (Node2Vec) Link Prediction, 10% masked | |
|---|---|
| `batch_size` | 64 (IMDB-B & ENZYMES), 32 (PROTEINS) |
| `optimizer` | Adam |
| `learning_rate` | 2e-4 |
| `dropout` | 0.2 |
| `hidden_dim` | 32 |
| `num_layers` | 5 |
| `num_linears` | 2 |
| `channels` | {c_init: 2, c_hid: 8, c_final: 4} |
| `train/val/test_noise_std` | 0.01(IMDB-B), 0.1 (PROTEINS & ENZYMES) |
| `ode_steps` (Euler, $K$) | 1 to 100 |
| `prior` | Node2Vec (per-graph classifier) |

| PIFM (Node2Vec) Link Prediction, 50% masked | |
|---|---|
| `batch_size` | 64 (IMDB-B & ENZYMES), 32 (PROTEINS) |
| `optimizer` | Adam |
| `learning_rate` | 2e-4 |
| `dropout` | 0.2 |
| `hidden_dim` | 32 |
| `num_layers` | 5 |
| `num_linears` | 2 |
| `channels` | {c_init: 2, c_hid: 8, c_final: 4} |
| `train/val/test_noise_std` | 0.01(IMDB-B), 0.1 (PROTEINS & ENZYMES) |
| `ode_steps` (Euler, $K$) | 1 to 100 |
| `prior` | Node2Vec (per-graph classifier) |

| Hyperparameter: PIFM(SIGL) | Value |
|---|---|
| `denoiser epochs` | 1000 |
| `SIGL hyperparams` | same as original paper |
| `optimizer` | Adam |
| `learning_rate` | 2e-4 |
| `dropout` | 0.2 |
| `hidden_dim` | 32 |
| `num_layers` | 5 |
| `num_linears` | 2 |
| `channels` | {c_init: 2, c_hid: 8, c_final: 4} |
| `train_noise_std` (masked $t$=0) | 0.05 |
| `val_noise_std` (masked $t$=0) | 0.05 |
| `ode_steps` (Euler, $K$) | 1000 (default) |
| `prior` | SIGL (pretrained graphon; `sort_ckpt`, `inr_ckpt`) |

**PIFM (GraphSAGE) Link Prediction, 10% masked**

| | |
|---|---|
| `batch_size` | 64 (IMDB-B & ENZYMES), 32 (PROTEINS) |
| `optimizer` | Adam |
| `learning_rate` | 2e-4 |
| `dropout` | 0.2 |
| `hidden_dim` | 32 |
| `num_layers` | 5 |
| `num_linears` | 2 |
| `channels` | {c_init: 2,  c_hid: 8,  c_final: 4} |
| `train/val/test_noise_std` | 0.05 (IMDB-B), 0.1 (PROTEINS & ENZYMES) |
| `ode_steps` (Euler, $K$) | 1 to 100 |
| `prior` | GraphSAGE (default hyperparameters) |

**PIFM (GraphSAGE) Link Prediction, 50% masked**

| | |
|---|---|
| `batch_size` | 64 (IMDB-B & ENZYMES), 32 (PROTEINS) |
| `optimizer` | Adam |
| `learning_rate` | 2e-4 |
| `dropout` | 0.2 |
| `hidden_dim` | 32 |
| `num_layers` | 5 |
| `num_linears` | 2 |
| `channels` | {c_init: 2,  c_hid: 8,  c_final: 4} |
| `train/val/test_noise_std` | 0.05 (IMDB-B & PROTEINS), 0.1 (ENZYMES) |
| `ode_steps` (Euler, $K$) | 1 to 100 |
| `prior` | GraphSAGE (default hyperparameters) |

| **Flow w/ Gaussian Prior** | **Value** |
|---|---|
| `epochs` | 1000 |
| `batch_size` | 64 (default), 32 (PROTEINS) |
| `optimizer` | Adam |
| `learning_rate` | 2e-4 |
| `dropout` | 0.2 |
| `hidden_dim` | 32 |
| `num_layers` | 5 |
| `num_linears` | 2 |
| `channels` | {c_init: 2,  c_hid: 8,  c_final: 4} |
| `train_noise_std` (masked $t{=}0$) | 0.00 |
| `val_noise_std` (masked $t{=}0$) | 0.00 |
| `ode_steps` (Euler, $K$) | 1 to 100 |
| `prior` | None (masked entries initialized from $\mathcal{N}(0.5, 1)$) |

| **DiGress + RePaint** | **Value** |
|---|---|
| `train.n_epochs` | 3000 |
| `train.batch_size` | 12 |
| `model.diffusion_steps` | 1000 |
| `model.n_layers` | 8 |
| `model.lambda_train` | [5, 0] |
| `model.extra_features` | all |
| `model.hidden_mlp_dims` | {X: 128, E: 64, y: 128} |
| `model.hidden_dims` | {dx: 256, de: 64, dy: 64, n_head: 8, dim_ffX: 256, dim_ffE: 64, dim_ffy: 256} |

| GDSS + RePaint | Setting |
|---|---|
| Sampler predictor / corrector | S4 / None |
| $n_{\text{steps}}$ / SNR / scale_eps | 1 / 0.15 / 0.7 |
| Probability flow / noise removal / $\varepsilon$ | False / True / $10^{-5}$ |
| Batch size (DataLoader) | from config (e.g., 12) |
| Mask mode (default) | `dataset` |
| Observed graph $A_{\text{obs}}$ | $(A_{\text{true}} \odot \text{mask})$; symmetrized, no self-loops |
| Binarization threshold (metrics) | 0.5 |

### E.4 METRICS CALCULATION

We evaluate performance only on the set of masked (unknown) edges in the upper triangle of the adjacency matrix. For each test graph, all metrics are computed on these entries and then averaged across graphs.

**Metrics Used in Tables** We report the following four metrics in the main results:

- **Area Under the ROC Curve (AUC).** Computed on the raw predicted scores (when available). AUC measures the probability that a randomly chosen true edge receives a higher predicted score than a randomly chosen non-edge. Larger AUC indicates stronger ranking performance.

- **Average Precision (AP).** Computed from the precision–recall curve induced by ranking the predictions. AP summarizes how well the model recovers true edges across all possible thresholds, with higher values indicating better precision–recall trade-offs.

- **False Positive Rate (FPR).** After thresholding predictions at $0.5$, the FPR is defined as

$$\text{FPR} = \frac{\text{FP}}{\text{FP} + \text{TN}},$$

- **False Negative Rate (FNR).** After thresholding predictions at $0.5$, the FNR is defined as

$$\text{FNR} = \frac{\text{FN}}{\text{FN} + \text{TP}},$$

- **MMD.** A kernel-based method that measures the difference between two probability distributions by embedding them in a feature space and finding the maximum difference between their mean embeddings.

AUC and AP are threshold-independent metrics (computed directly on the provided scores), while FPR and FNR are threshold-dependent error rates (obtained after binarizing at $0.5$). All values reported in the tables are averaged over test graphs and expressed in percentage.

## F ADDITIONAL RESULTS

### F.1 DENOISING.

This second problem is the complement of expansion, meaning that we seek to remove a set of spurious edges $\mathcal{E}_S$ from $A^{\mathcal{O}}$, such that the edge set of the ground truth is $\mathcal{E} = \mathcal{E}_O \setminus \mathcal{E}_S$. Hence, the initialization becomes $\mathbf{A}_0 = \mathbf{A}_1^{\mathcal{O}} \odot (f_{prior}(\mathbf{A}_1^{\mathcal{O}}) + \boldsymbol{\epsilon}_s)$. We assume that 20% of the edges are flipped; the results are shown in Table 5. Similarly to expansion, PIFM (GraphSAGE) attains the best AUC/AP on all datasets, again surpassing the GraphSAGE prior and remaining baselines. It *strongly* reduces false positives from the given prior initialization, while FNR is low on dense IMDB-B (2.67) and higher on sparser sets. Overall, PIFM removes spurious edges more reliably while improves other metrics as well.

Table 5: Performance for the **denoising** task with **20% of upper-triangle 0-entries flipped (0.2 Flip)**. We report AUC, Average Precision (AP↑), False Positive Rate (FPR↓), and False Negative Rate (FNR↓), all in percent (%). The best result for each metric is in **bold blue** and the second best is green. The metrics are restricted on the upper-triangle 1-region of $A^{\mathcal{O}}$, and compared against $\mathbf{A}_1$ on that region.

| | | | | | | | | | | | | |
|---|---|---|---|---|---|---|---|---|---|---|---|---|
| **Flip Rate: 20% (0.2 Flip)** | | | | | | | | | | | | |
| | **ENZYMES** | | | | **PROTEINS** | | | | **IMDB-B** | | | |
| Method | AP↑ | AUC↑ | FNR↓ | FPR↓ | AP↑ | AUC↑ | FNR↓ | FPR↓ | AP↑ | AUC↑ | FNR↓ | FPR↓ |
| *Baselines* | | | | | | | | | | | | |
| GraphSAGE | 68.19 | 73.89 | **16.14** | 61.72 | 73.79 | 76.70 | **12.68** | 60.47 | 92.54 | 77.29 | 16.77 | 53.37 |
| DiGress + RePaint | 41.98 | 49.38 | 87.91 | **13.33** | 49.36 | 51.20 | 78.44 | **18.19** | 80.54 | 51.59 | 73.41 | 23.49 |
| GDSS + RePaint | 44.59 | 50.86 | 69.18 | 29.70 | 49.72 | 49.43 | 68.39 | 32.64 | 82.36 | 53.23 | 69.05 | 26.00 |
| Flow w/ Gaussian prior | 49.90 | 54.56 | 52.30 | 38.93 | 57.18 | 58.70 | 62.84 | 24.32 | 96.75 | 94.63 | 3.80 | 12.66 |
| *Ours* | | | | | | | | | | | | |
| PIFM (GraphSAGE) | **69.40** | **77.17** | 45.66 | 18.14 | **77.43** | **81.78** | 32.41 | 20.91 | **98.46** | **96.52** | **2.67** | **12.10** |

## F.2 ADDITIONAL EXPERIMENTS

Our method has two main hyperparameters:

1. $\sigma_s$, which is used for computing $\boldsymbol{\epsilon}_s \sim \mathcal{N}(0, sigma_s^2)$ in $\mathbf{A}_0 = \xi \odot \mathbf{A}_1 + (1 - \xi) \odot \left( f_{\text{prior}}(\mathbf{A}_1^{\mathcal{O}}) + \boldsymbol{\epsilon}_s \right)$

2. $K$, which are the total number of steps in the Euler approximation

### F.2.1 PERFORMANCE AS A FUNCTION OF $\sigma_s$

We run an ablation of the performance of PIFM with GraphSAGE as a function of $\sigma_s$. We focus on ENZYMES and IMDB, and we evaluate the ROC for the best value of $K$ for each noise level.

The ablation is illustrated in Fig. 4. First, we observe that the gains of using PIFM are higher for a smaller drop rate, as expected; in particular, we observe that PIFM with $\sigma_s$ jumps from $\approx 0.73$ for $\sigma_s = 0$ to $\approx 0.81$ for $\sigma_s = 0.1$. Second, for both configurations, performance peaks not at zero noise, but at a small noise level of $\sigma_s = 0.1$. This suggests that a slight injection of noise benefits model generalization. Beyond this optimal point, increasing the noise level leads to a steady decline in performance, meaning that the effect of the prior decreases, as expected.

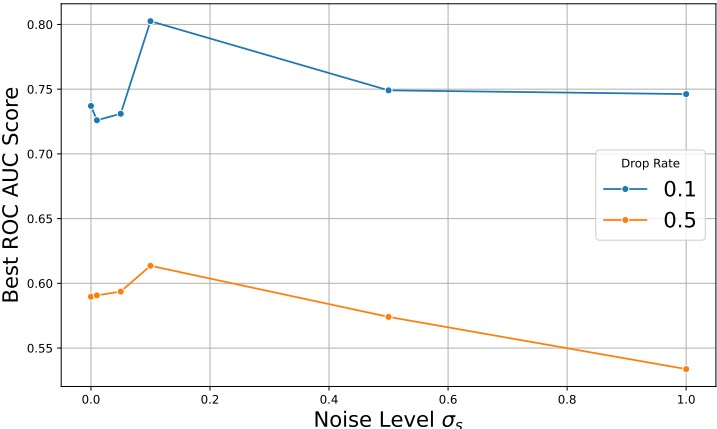

Figure 4: ROC as a function of the noise $\sigma_s$ in $p(\mathbf{A}_0)$. The impact of noise level $\sigma_s$ on model performance, measured by the best ROC AUC score. Results are shown for two different drop rates: 0.1 (blue) and 0.5 (orange). A small amount of noise improves performance for both configurations, after which increasing noise leads to performance degradation.

### F.2.2 PERFORMANCE AS A FUNCTION OF $K$

To determine the optimal number of processing steps, $K$, we evaluated model performance while varying this parameter from 1 to 100. Figures 5 and 6 shows the results for a fixed drop rate of 0.1 and 0.5 respectively, across five different noise levels.

A key observation is that peak performance, in terms of AUC-ROC, is achieved within a very small number of steps, typically for $K < 10$. In particular, the introduction of a moderate noise level allows the model to achieve its highest overall score ($\approx 0.80$ ROC AUC) in a single step ($K = 1$). However, this advantage diminishes as the number of steps increases. The model without noise ($\sigma_s = 0.0$) provides the most stable and consistently high performance for larger $K$. Conversely, a high noise level ($\sigma_s = 1$) consistently degrades performance regardless of the number of steps. This analysis suggests a trade-off: while noise can provide a significant boost for models with very few steps, a no-noise configuration is more robust for models with a larger number of steps.

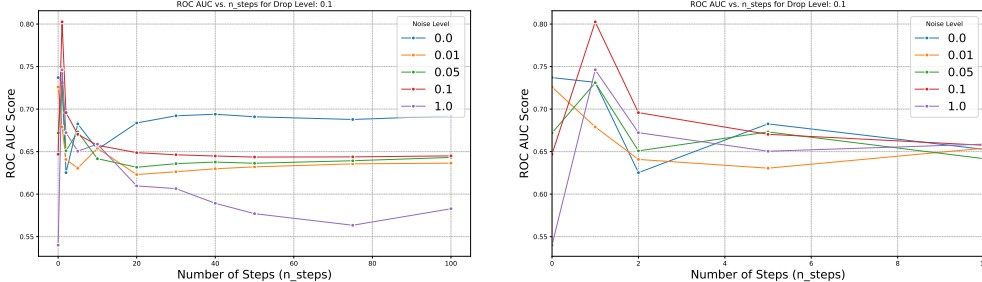

Figure 5: An analysis of the ROC AUC score as a function of the number of processing steps (K) for a drop rate of 10%. This experiment was conducted with a fixed drop rate of 0.1, while varying the noise level, $\sigma_s$. The results show that the optimal number of steps is small, typically under 10.

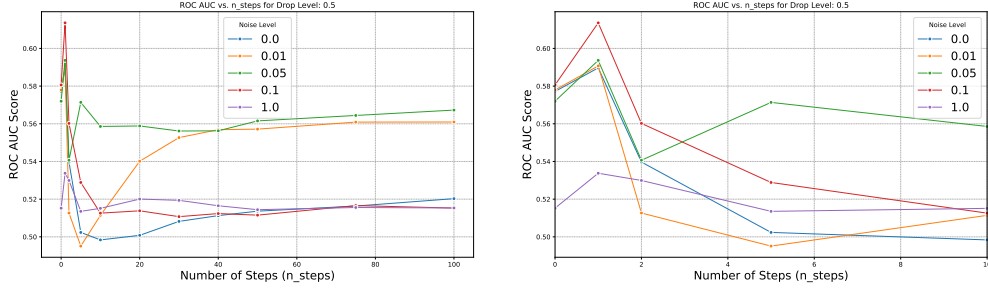

Figure 6: An analysis of the ROC AUC score as a function of the number of processing steps (K) for a drop rate of 50%. This experiment was conducted with a fixed drop rate of 0.1, while varying the noise level, $\sigma_s$. The results show that the optimal number of steps is small, typically under 10.

### F.3 DISTORTION-PERCEPTION TRADE-OFF.

Here we expand on the distortion-perception trade-off by computing the MMD. The results are shown in Figures 7 and 8. Again, both figures show that the MMD² distance decreases as we increase $K$; this is particular noticeable for $0 < \sigma_s \leq 0.1$. In other words, if we aim for a high-quality perceptual reconstruction, we should consider $\sigma_s = 0.01$ or $0.05$. However, if we are aiming for high reconstruction quality in terms of AUC-ROC, we should use $\sigma_s = 0.1$ (see Fig.4). In other words, the choice of the $K$ is heavily dependent of the downstream task.

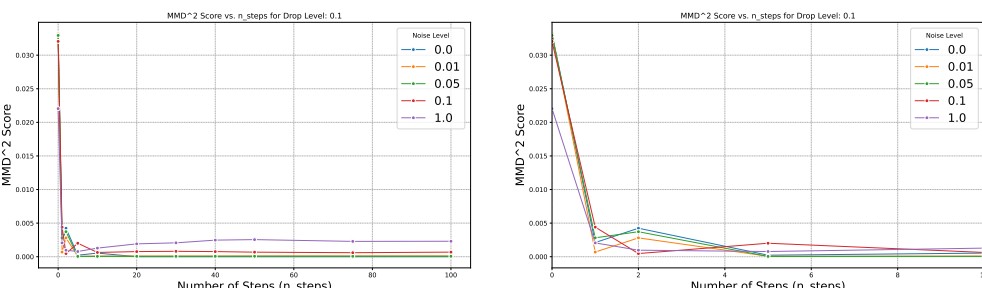

Figure 7: Analysis of the perception component of the distortion-perception trade-off. The plot shows the MMD² score (where lower is better) versus the number of steps, K, for a fixed drop rate of 0.1. Each line represents a different noise level $\sigma_s$.

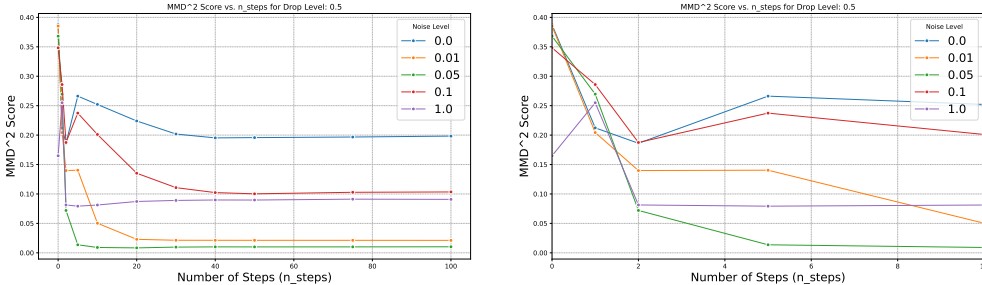

Figure 8: Analysis of the perception component of the distortion-perception trade-off. The plot shows the MMD² score (where lower is better) versus the number of steps, K, for a fixed drop rate of 0.5. Each line represents a different noise level $\sigma_s$.

### F.3.1 ADDITIONAL METRICS

This section incorporates additional metrics to showcase the observed performance trade-off. The resultsa for IMDB are in Figs. 9 and 10, for PROTEINS in Figs. 11 and 12, and for ENZYMES in Figs. 13 and 14. While an increased number of steps yields an improvement in generating an estimated graph with statistics that more closely align with the ground-truth distribution, the reconstruction performance (measured in terms of AUC) declines relative to the initial step. Critically, the trend is found to be highly contingent on the underlying dataset's sparsity. For the dense case (IMDB), the AUC exhibits a consistent monotonic decrease after the optimal initial guess, independent of drop rates. In contrast, the sparser PROTEINS and ENZYMES datasets demonstrate an intermediate improvement as the number of steps increases, though their overall AUC still trails that achieved at the initial step ($t = 1$).

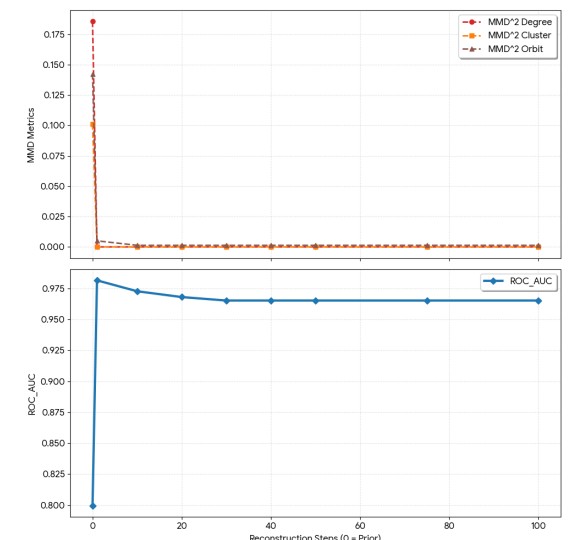

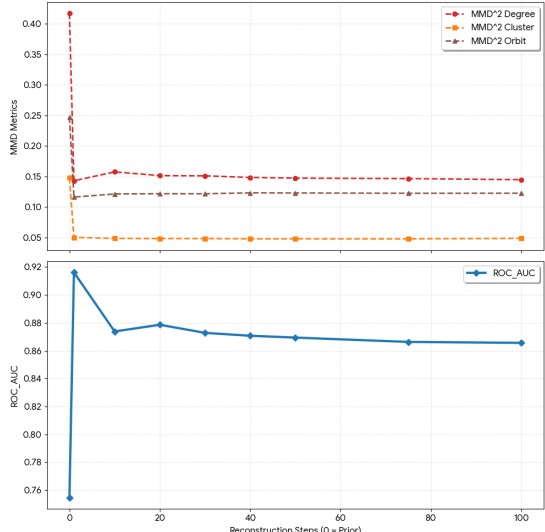

Figure 9: IMDB dataset, expansion task, 10% drop rate

Figure 10: IMDB dataset, expansion task, 50% drop rate

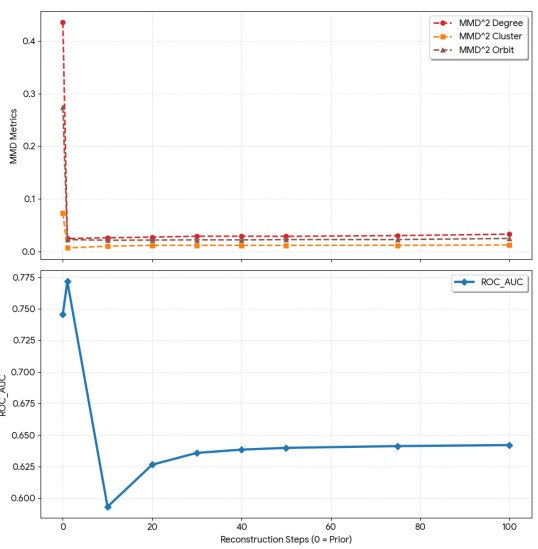

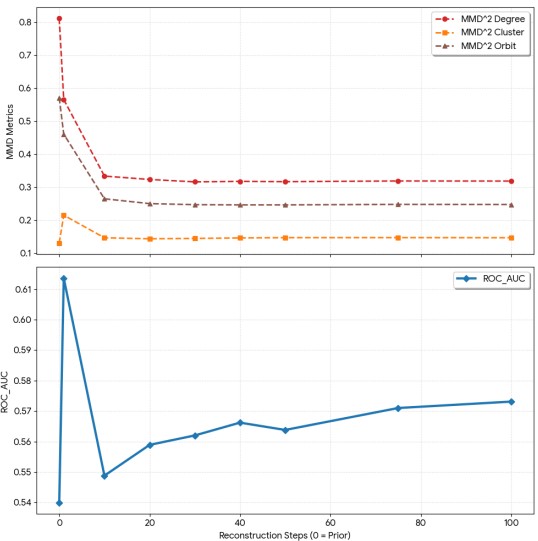

Figure 11: PROTEINS dataset, expansion, 50% drop rate

Figure 12: PROTEINS dataset, expansion, 50% drop rate

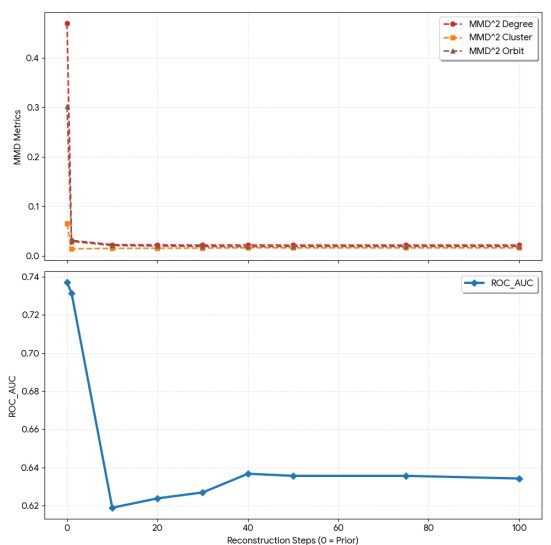

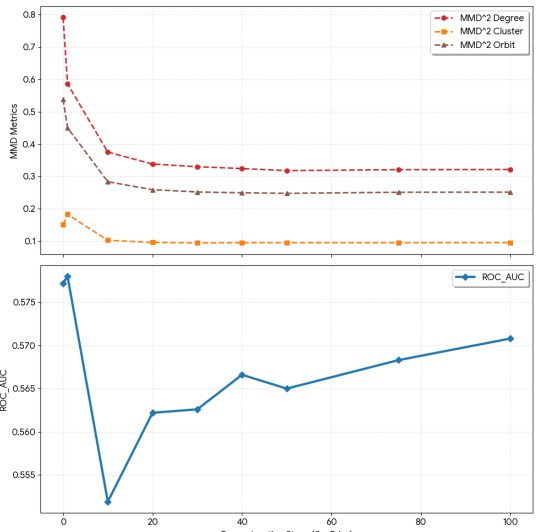

Figure 13: ENZYMES dataset, expansion, 10% drop rate

Figure 14: ENZYMES dataset, expansion, 50% drop rate

## F.4 EXAMPLES OF RECONSTRUCTED GRAPHS

We show here a few samples for the expansion case. We plot the samples from ENZYMES, using a subset of the dataset used in Section 5.3.

**Binary comparison.** In this case, we first compare the thresholded versions (with 0.5) of the mean matrices. We compute this for 3 graphs in the test set (3, 7 and 29). The plots are in Figs. 15, 16 and 17

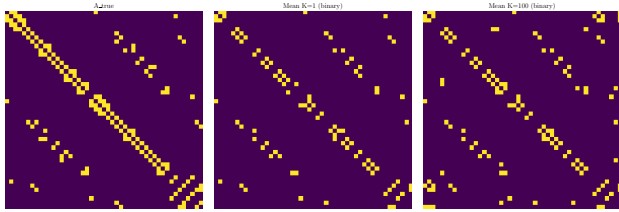

Figure 15: Graph reconstruction for sample 3, thresholded with 0.5

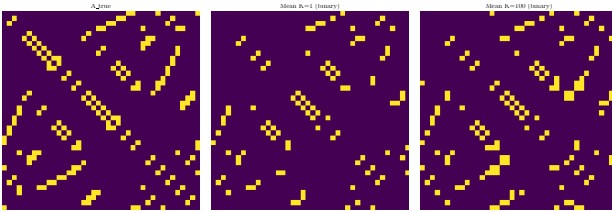

Figure 16: Graph reconstruction for sample 7, thresholded with 0.5

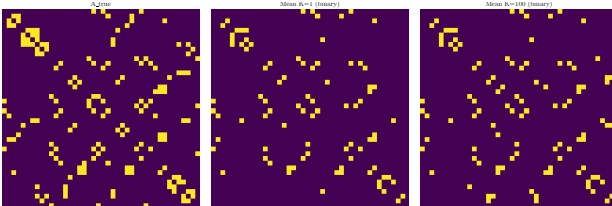

Figure 17: Graph reconstruction for sample 30, thresholded with 0.5

**Raw comparison - Mean.** In this case, we compare the raw versions of the mean matrices. We compute this for 3 graphs in the test set (3, 7 and 29). The plots are in Figs. 18, 19 and 20. Notice that the mean reconstructions for $K = 100$ have values that are between 0 and 1; this can be explained by looking at individual samples (see below), which are more diverse, and therefore, they have non-overlapping set of existing edges.

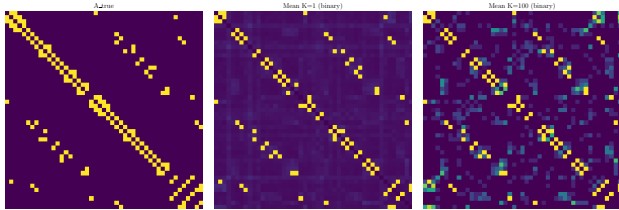

Figure 18: Graph reconstruction for sample 3, mean raw

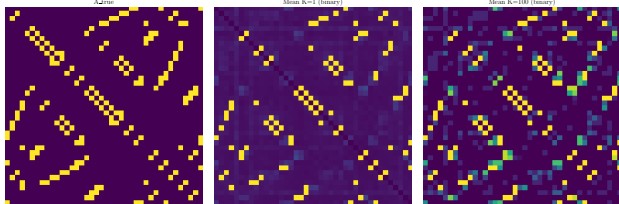

Figure 19: Graph reconstruction for sample 7, mean raw

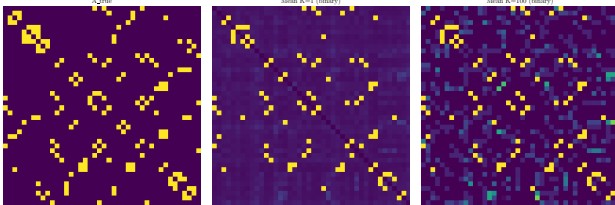

Figure 20: Graph reconstruction for sample 30, mean raw

**Raw comparison - Median.** In this case, we compare the raw versions of the median matrices. We compute this for 3 graphs in the test set (3, 7 and 29). The plots are in Figs. 21, 22 and 23.

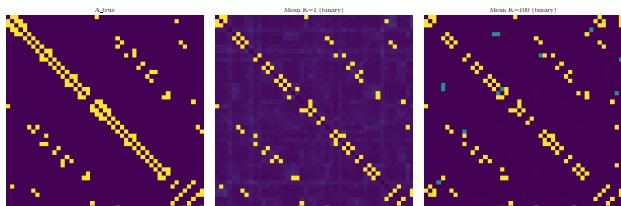

Figure 21: Graph reconstruction for sample 3, median raw

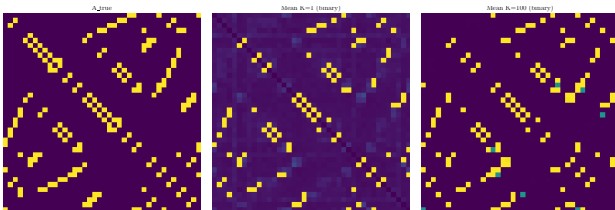

Figure 22: Graph reconstruction for sample 7, median raw

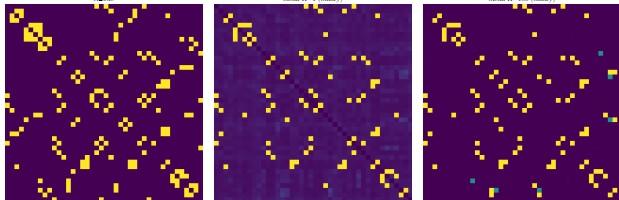

Figure 23: Graph reconstruction for sample 30, median raw

**Individual samples for each graph.** Lastly, we show the raw versions of different realizations (individual samples) for each graph. We compute this for 3 graphs in the test set (3, 7 and 29). Interesting, the samples for $K = 100$ are more diverse (similar to the case of images in Ohayon et al. (2025)); this diversity explains why the raw mean in Figs. 18, 19 and 20 have values that are not exactly 0 or 1 (which means that there are non-overal between samples).



Figure 24: Individual samples for $K = 1$ and for sample 3

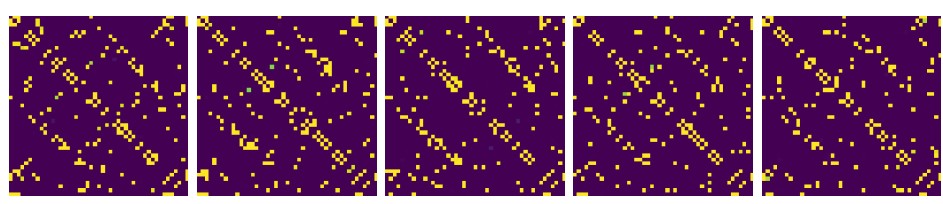

Figure 25: Individual samples for $K = 100$ and for sample 3

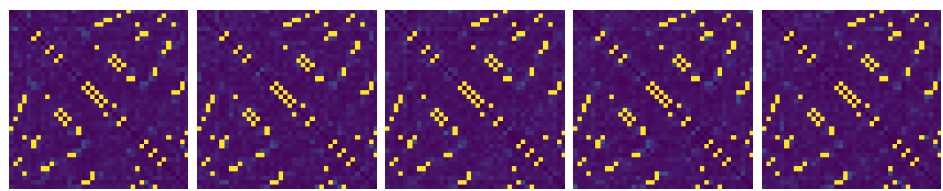

Figure 26: Individual samples for $K = 1$ and for sample 7

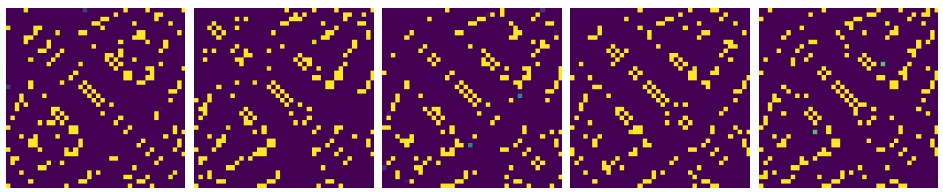

Figure 27: Individual samples for $K = 100$ and for sample 7



Figure 28: Individual samples for $K = 1$ and for sample 30

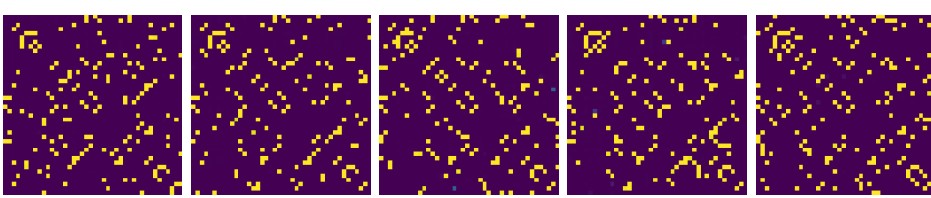

Figure 29: Individual samples for $K = 100$ and for sample 30

### F.5 PIFM ON LARGE-SCALE GRAPHS

In this section, we train PIFM on large-scale graphs. In particular, we focus on CORA Yang et al. (2016).

**Building the datataset.** To enable scalable diffusion training on Cora while maintaining full-graph link prediction capabilities, we introduce a subgraph-based variant of PIFM following Limnios et al. (2023). We instantiate this via an edge-centered sampling scheme, where each subgraph represents a k-hop ego-network (capped at a maximum node count) centered around a seed edge from the training split. To ensure reproducibility, we sample a fixed set of seed edges and corresponding subgraphs which remain constant throughout each run.

Following a standard 85/5/10 (train/validation/test) edge split, we evaluate link prediction on held-out edges using the protocol established in the main paper. Furthermore, we utilize purely structural node features. By default, we concatenate: (i) Laplacian positional encodings derived from the smallest $k$ generalized eigenvectors of $Lv = \lambda Dv$ computed on the training adjacency; and (ii) a 2-dimensional local context vector comprising both raw and normalized node degrees.

**Training.** To initialize PIFM, we first pre-train structural priors (GraphSAGE and Node2Vec) on the full Cora graph using only the training edge split (85% of edges). These learned embeddings are subsequently used to initialize PIFM for inference on the test split (10% of edges).

During PIFM training, each training edge seeds a unique edge-centered subgraph. Within each subgraph, we define the observed context as all other training edges, and a hidden region comprising all remaining node pairs (including non-edges and edges outside the context). The seed edge is explicitly masked from the context and designated as the sole supervision target. Consequently, each training example tasks the model with reconstructing a single missing edge within its local neighborhood.

We construct the initial state matrix $\mathbf{A}_0$ by combining the observed context with structural prior predictions in the hidden region. The flow model is then trained to denoise $\mathbf{A}_0$ toward the ground-truth local adjacency. While the flow ODE updates all entries within the hidden region, the gradient is computed exclusively from the seed edge. This adapts the global training procedure of Algorithm 1 to a localized, subgraph-based regime

**Inference.** For inference, every held-out validation or test edge seeds a k-hop subgraph. We define the observed context using the training edges, remove the centered test edge from the mask, and apply diffusion to the resulting hidden region. We strictly evaluate the prediction for the centered held-out edge in each subgraph. To resolve potential overlapping, we aggregate predictions via logit averaging across all subgraphs where a specific edge is present, producing a single probability matrix over all node pairs. We then compute metrics on the held-out positive and negative edges. Since PIFM and the structural prior baseline are evaluated on the exact same set of pairs, the results isolate the specific benefits of the diffusion process

Table 6: Link prediction on CORA (hidden edges) with a GraphSAGE structural prior. We report AUC, Average Precision (AP), False Positive Rate (FPR), False Negative Rate (FNR), and Mean Reciprocal Rank (MRR). We report PIFM's performance on Cora for different $k$-hop neighborhoods during subgraph sampling, and the embedding dimension of the laplacian positional encodings. We include also the average node counts for each subgraph for each case.

| Method | AUC↑ | AP↑ | FPR↓ | FNR↓ |
|---|---|---|---|---|
| GraphSAGE | 95.61 | 47.76 | 0.59 | 23.79 |
| 1hop 2dim-embed ($\sim$ 16.7 nodes) | 87.47 | 61.07 | 1.00 | 0.00 |
| 2hop 2dim-embed ($\sim$ 58.0 nodes) | 96.25 | 75.39 | 1.00 | 0.00 |
| 3hop 2dim-embed ($\sim$ 154.5 nodes) | **97.03** | 33.71 | 1.00 | 0.00 |
| 4hop 8dim-embed ($\sim$ 210.4 nodes) | 96.14 | 21.53 | 1.00 | 0.00 |
| NCNC Wang et al. (2024) (from Li et al. (2023) | 96.90 | - | - | - |
| NCN Wang et al. (2024) (from Li et al. (2023) | 96.76 | - | - | - |

**Results.** Results in Table 6 show that PIFM is learning useful global structural representations even on very large graphs by exploiting the subgraph sampling approach. Increasing the neighborhood size is also helpful for PIFM to obtain a richer set of local information on large graphs like CORA. We compare with a few baselines from Li et al. (2023); we used their reported results. In addition to the metrics, we have included Figure 31 and Figure 32 for visualizations of the intermediate adjacencies of PIFM when reconstructing on CORA.

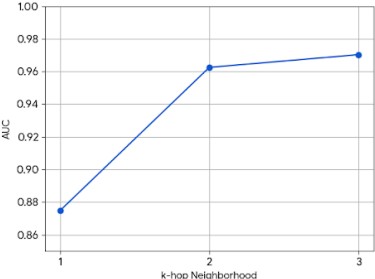

Figure 30: AUC vs $k$-hop neighborhood. We observe that adding more hops enhances the performance of PIFM, as it exploits more information.

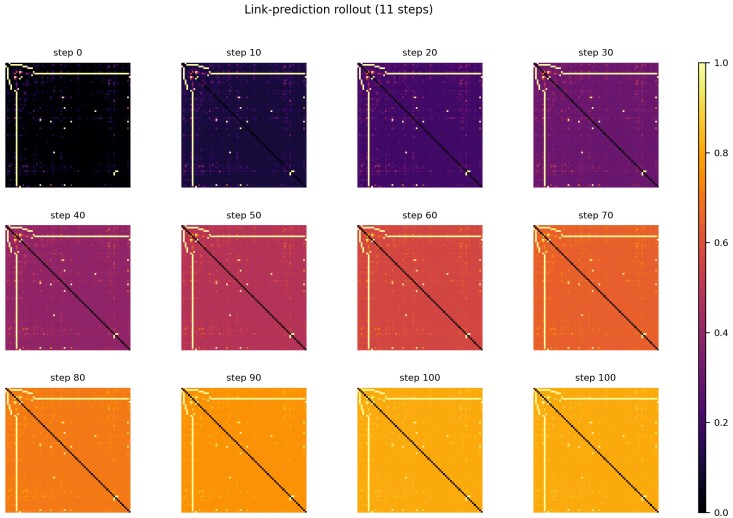

Figure 31: Visualization of the intermediate adjacency matrices of a sugraph for CORA.

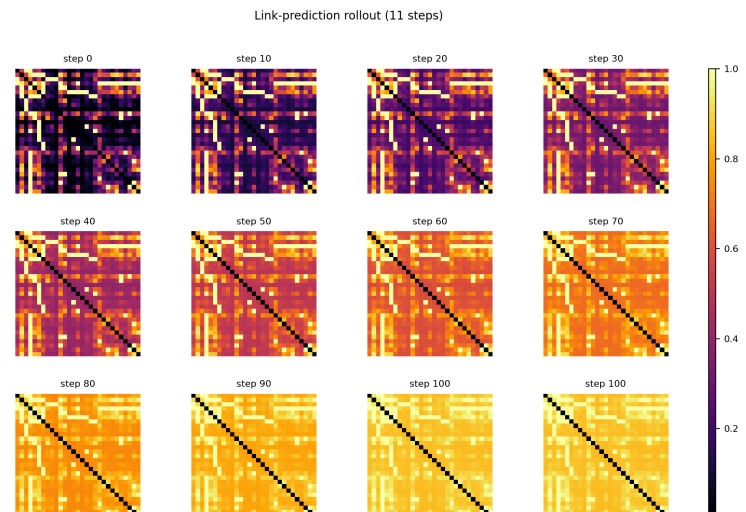

Figure 32: Visualization of the intermediate adjacency matrices of a sugraph for CORA.

## F.6 INTERMEDIATE ADJACENCY MATRICES

In Figs. 33- 35, we show visualizations of the diffusion trajectory of the link-prediction sampler by "snapshotting" the predicted adjacency matrix at steps 0 to 100 in 10-step-increment of a 100-step-total sample path. Each panel shows the raw adjacency values, zeroed on the diagonal, rendered with the colormap with black associated to 0 and yellow to 1. The most bottom-right panel is the ground-truth adjacency for comparison.

To see the sampling process, progress from left-to-right and top-to-bottom shows how the sampler denoises toward the final reconstruction (steps=100). From those images we could see the smooth transitions along the full reconstruction trajectory of PIFM.

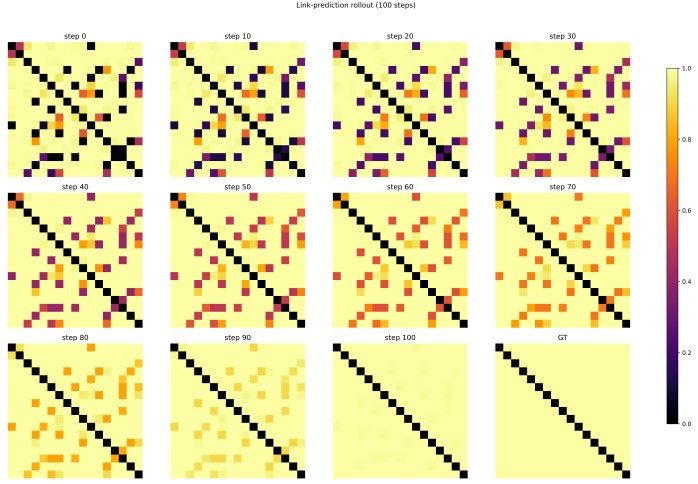

Figure 33: Visualization of IMDB 50% drop rate reconstruction. (Graph 1)

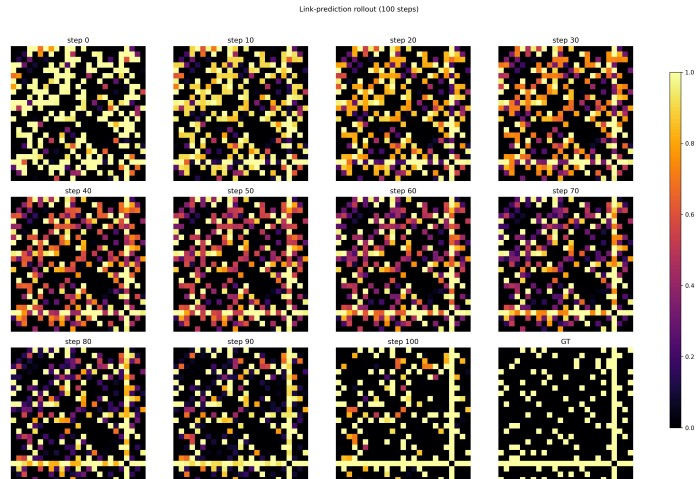

Figure 34: Visualization of IMDB 50% drop rate reconstruction. (Graph 2)

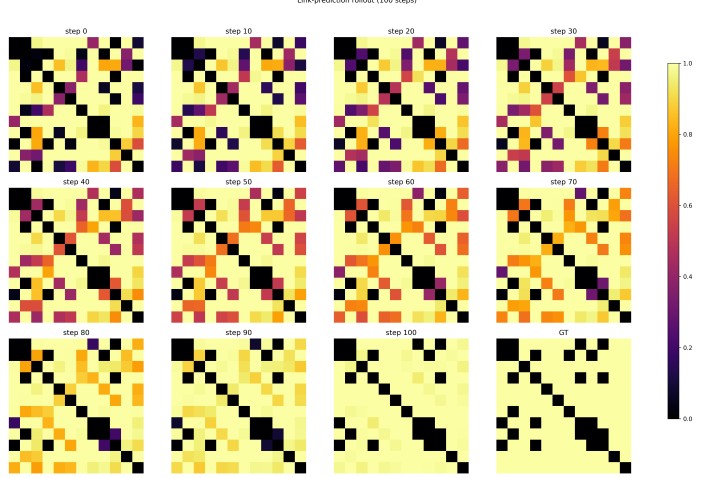

Figure 35: Visualization of IMDB 50% drop rate reconstruction. (Graph 3)

## F.7 TRAINING COST

All the models are trained on NVIDIA A100-SXM4-80GB GPUs. Fitting the Node2Vec prior takes roughly 1 to 1.5 hours depending on the datasets. Once the prior is fixed, the flow model itself has essentially the same training loop and cost as when using a Gaussian prior: end-to-end training of the flow model itself takes about 2 hours on ENZYMES and IMDB-B, and about 4 to 5 hours on PROTEINS. We did not observe a noticeable difference in training behavior between the Gaussian and prior-informed variants, and the model behaves similarly under different edge-drop rates.

## F.8 TRANSFERABILITY

We have experimented the 10% and 50%-drop-rate IMDB-B checkpoints for PIFM and used on PROTEINS and ENZYMES dataset with the same drop rates, and the best results and the results at the end of the 100 sampling steps of each run are as shown below.

Table 7: Transferability of IMDB-B PIFM checkpoints (with graphSAGE priors) to PROTEINS and ENZYMES. We use the 10% and 50%-drop-rate checkpoints trained on IMDB-B and evaluate them on the corresponding 10% and 50% drop-rate settings of the target datasets. For each setting, we report the best value over 100 sampling steps and the value at the final step ($t = 100$). Metrics are Average Precision (AP), AUC, False Negative Rate (FNR), and False Positive Rate (FPR). All metrics are reported in percent.

| Source checkpoint | Target dataset | AP↑ | AUC↑ | FNR↓ | FPR↓ |
|---|---|---|---|---|---|
| *Structural priors (no flow)* | | | | | |
| PROTEINS(10% drop) | ENZYMES (10% drop) | 41.28 | 73.70 | 13.49 | 60.59 |
| IMDB-B (10% drop) | PROTEINS (10% drop) | 46.36 | 74.58 | 11.00 | 63.50 |
| ENZYMES (10% drop) | PROTEINS (10% drop) | 46.36 | 74.58 | 11.00 | 63.50 |
| PROTEINS (50% drop) | ENZYMES (50% drop) | 23.08 | 57.72 | 40.02 | 52.16 |
| IMDB-B (50% drop) | PROTEINS (50% drop) | 27.71 | 53.99 | 32.16 | 66.86 |
| ENZYMES (50% drop) | PROTEINS (50% drop) | 27.71 | 53.99 | 32.16 | 66.86 |
| *PIFM* | | | | | |
| PROTEINS (10% drop) | ENZYMES (10% drop) | 43.87 | 76.96 | 71.96 | 4.24 |
| IMDB-B (10% drop) | PROTEINS (10% drop) | 46.79 | 75.53 | 47.07 | 11.62 |
| ENZYMES (10% drop) | PROTEINS (10% drop) | 49.63 | 76.60 | 59.02 | 6.63 |
| PROTEINS (50% drop) | ENZYMES (50% drop) | 25.68 | 61.88 | 95.37 | 1.70 |
| IMDB-B (50% drop) | PROTEINS (50% drop) | 24.88 | 59.02 | 61.02 | 24.32 |
| ENZYMES (50% drop) | PROTEINS (50% drop) | 26.81 | 55.80 | 86.13 | 9.97 |

From the table we can see that the model degrades the predictions a lot at lower (10%) drop rates, making the results incomparable when comparing to results in Table 1. It can be seen that with more steps integrated, at the final step $t = 100$ the metrics are much worse than the best result over all sampling steps or the structural prior. Although at higher (50%) drop rates the best results are more comparable to the best results in Table 2, this is due to the graphs being very corrupted under 50% drop rate and hence the observed graphs are much more degraded than they used to be.

## F.9 THRESHOLDING LEVELS

We investigated the impact of alternative thresholding levels—specifically 0.3 and 0.7, beyond the standard 0.5—on the evaluation metrics, with results presented in Figures 36 and 37. We observed that adjusting the threshold yields a marginal increase in the Area Under the Curve (AUC). We hypothesize this is partly driven by the model's inherent behavior of actively pushing output values towards the binary extremes (0 or 1). However, we noted a slightly larger, albeit still minimal, improvement for lower values of K, where the influence of the prior is greater and output values are typically further away from 0 and 1. Crucially, despite these dynamics, the use of an alternative threshold does not lead to a statistically significant improvement in overall performance.

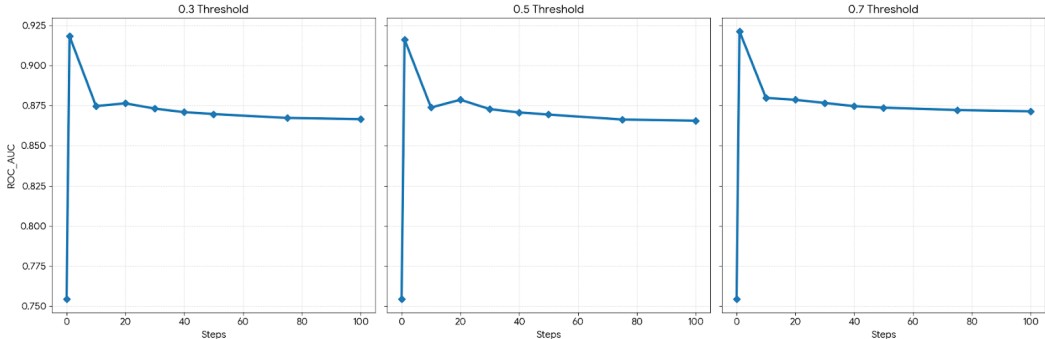

Figure 36: ROC-AUC vs number of steps sampled for IMDB dataset, link prediction task, 50% drop rate under 0.3, 0.5, and 0.7 thresholding levels.

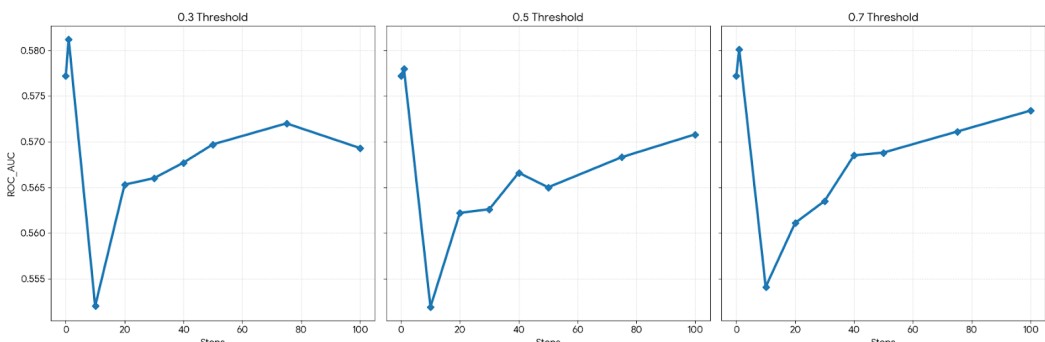

Figure 37: ROC-AUC vs number of steps sampled for ENZYMES dataset, link prediction task, 50% drop rate under 0.3, 0.5, and 0.7 thresholding levels.

## G    LIMITATIONS

Despite these advantages, PIFM has apparent limitations. It's performance heavily depends on the quality of prior estimation, as shown by the gap between the node2vec-prior and graphon-prior versions of PIFM. Moreover, graphons may not be the most suitable prior in practice: they are fundamentally limit objects defined for limits of dense graphs, which restricts their applicability to sparse real-world networks. Graphons also does not capture dependencies between edges beyond what can be explained through the latent coordinates. Additionally, the current formulation is restricted to undirected and unweighted graphs, and the training overhead is higher than one-shot priors.

Looking forward, promising directions include extending PIFM to incorporate node and edge attributes for richer graph inference tasks, scaling the method to larger and more complex real-world networks, and enhancing the graphon prior by learning a dictionary of graphons from which the model can adaptively select during sampling. Such a design would provide more faithful prior initialization for datasets containing diverse graph types.

