# OpenReview forum: "Prior-Informed Flow Matching for Graph Reconstruction"
_ICLR.cc/2026/Conference — Submitted to ICLR 2026_

### Official Review · Reviewer_vUVC · 2025-10-19

**Soundness:** 3
**Presentation:** 3
**Contribution:** 2
**Rating:** 4
**Confidence:** 4

**Summary:**

The paper proposes Prior-Informed Flow Matching (PIFM), a conditional flow model for graph reconstruction. The authors reframe the graph reconstruction process as a two-stage method, grounded in distortion-perception theory. First, it uses a prior to form an informed initial estimate of the adjacency matrix, which serves as an approximation of the Minimum Mean Squared Error estimator to minimize distortion. Second, it applies rectified flow matching to refine this estimate, transporting it toward the true distribution of clean graphs by learning a global structural coupling to enhance perceptual quality . Experiments show competitive results on three datasets.

**Strengths:**

1. The framework of the paper is novel. Combining traditional graph embedding method with flow model is a good try, bridging the gap between local estimator and global estimator.

2. The paper is clear and easy to follow.

**Weaknesses:**

1. The experiments lack some strong baselines for link prediction. The authors mainly compare PIFM with its own prior, which is fine and, in my view, justifies the effectiveness of the flow matching technique. However, the absence of SOTA link prediction methods makes the experiments less persuasive.

2. The scale of the datasets is too limited and not representative. As stated by the authors, the model inherits the scalability challenges of diffusion models, and the selected datasets are too small for the task. It would be better to expand the size of the datasets and include various types, such as molecule datasets, bioinformatics datasets, social network datasets, etc.

**Questions:**

1. Utilizing prior knowledge for graph reconstruction is a good point, but prior knowledge is not always ready to use. I wonder when prior knowledge is required and can significantly improve the model?

2. For the graph embedding method selection, what's the criterion? Can we incorporate other GNN methods into the model?

---

> ### Author Response · Authors · 2025-11-25
>
> We sincerely thank the reviewer for the feedback. Below we reply to all the raised comments.
>
> ## **W.1 and 2; Lack of strong baseline for link prediction and scale of datasets**
>
> Given that link prediction methods are in general transductive, we added a new experiment showing that PIFM can be extended to large scale graphs.
> In particular, we train on CORA (see response to reviewer a7bB).
> For this case, we could add the methods from typical link prediction, where we used the top ones reported in [1] (NCNC and NCN). We hope this helps to show the merit of our proposed method.
>
> ## **Q.1 Importance of the prior**
>
> We thank the reviewer for raising this point.
> The importance of the prior strongly depends on the difficulty of the problem, where we measure this via the drop rate %.
> When looking at table 1 and 2, we consistently observe that for datasets like IMDB, the performance difference between approaches using a simple Gaussian prior (Flow with Gaussian prior) and our method (PIFM), which leverages structural prior knowledge, is significantly higher for a 50% drop rate compared to a 10% drop rate.
>
> This observation is consistent with findings across many other inverse problems in different scientific and engineering domains: when the problem is more ill-posed (meaning, there is less observed data available to constrain the solution), the role of the prior is fundamentally more critical and leads to greater performance gains.
>
> Therefore, prior knowledge is most important, and can significantly improve the model, when the number of missing edges is higher, resulting in a more ill-posed graph reconstruction task.
>
> Regarding the difference between various models used to generate the prior, we find that when a more powerful and expressive model is used to learn the initial representation, PIFM performs better. Specifically, in all our experiments, PIFM utilizing a GraphSAGE-based prior consistently outperforms PIFM utilizing a simpler node2vec-based prior.
>
>
> ## **Q.2 Criteria for the graph embedding method in the prior**
>
>
> We thank the reviewer for raising this point. In fact, we can incorporate any other GNN based method as long as it satisfies our two assumptions; this happens to almost all existing link prediction methods.
> Extending our formulation to more general cases, where these assumptions do not hold, is something we are working on and is out of the scope of our current paper.
>
>
>
> [1] Li, Juanhui, Harry Shomer, Haitao Mao, Shenglai Zeng, Yao Ma, Neil Shah, Jiliang Tang, and Dawei Yin. "Evaluating graph neural networks for link prediction: Current pitfalls and new benchmarking." Advances in Neural Information Processing Systems 36 (2023): 3853-3866.

---

### Official Review · Reviewer_cqqu · 2025-10-30

**Soundness:** 3
**Presentation:** 2
**Contribution:** 2
**Rating:** 4
**Confidence:** 3

**Summary:**

This paper proposed mainly a method for graph reconstructino with 2 steps:
1. initial graph reconstruction
2. refinement via optimal transport

**Strengths:**

The method makes sense to me in a high level way. It resembles a two-stage refinement process: the first stage performs a probability-based estimation, while the second stage learns an actual mapping from the imprecise reconstruction to the real graph.

**Weaknesses:**

I am not very clear about the motivation about this task or the motivation of the proposed method. See questions.

**Questions:**

There are several points that are confusing to me:

1. I do not fully understand the difference between graph reconstruction and graph generation. In graph generation, there are already ways to make the generated graphs very similar to the training data by leveraging priors that encourage overfitting toward the training distribution. With my understanding, using such a model, one could directly achieve graph reconstruction through an inpainting-like process. Is that correct?

2. For the first-stage graph probability estimation, I am confused why the method needs to rely on either graphon or GraphSAGE, for the following reasons:
* Graphon assumes that graphs of different sizes follow the same underlying distribution, which is often not true in real-world datasets. For example, once molecular size increases, this assumption can break.
* GraphSAGE predicts link existence using node embeddings, which is known to be less effective than graph transformers for such tasks. Instead, a graph generative model could also provide a probabilistic estimation.

---

> ### Author Response · Authors · 2025-11-25
>
> We thank the reviewer for the feedback, and we apologize for the lack of clarity. Please find below our replies to your questions.
>
>
> ## **W.1 Lack of clarity in the motivation**
>
> We appreciate the reviewer’s comment and apologize for the lack of clarity regarding our motivation. The goal of PIFM is to develop a global link prediction method, in contrast to classical approaches such as node2vec or GraphSAGE that rely on conditionally independent predictions in an embedding space. These local methods capture only neighborhood-level information and thus cannot model the global structural dependencies present in real networks.
>
> To address this, we first formulate link prediction as an inverse problem given partial observations of the graph. This formulation reveals a family of valid reconstructions depending on the reconstruction objective, and these solutions exhibit a trade-off that we formalize using a permutation-equivariant variant of the perception–distortion trade-off. Crucially, this characterization allows us to identify a canonical global estimator by focusing on the case $P=0$.
>
> In this setting, the optimal estimator is given by an optimal transport (OT) map between the MMSE estimator and the ground-truth graph distribution. The MMSE estimator, however, is generally local because its practical approximations rely on standard link prediction models that assume conditional independence.
>
> To obtain a global estimator, we therefore
>
> 1. Approximate the conditional mean (MMSE) using classical link prediction models; this yields a local but tractable baseline.
>
> 2. Learn the optimal transport map between this local MMSE estimate and the ground-truth graphs using a rectified flow model, which provides an efficient way to approximate OT in high dimensions.
>
> This combination allows PIFM to leverage existing local predictors while upgrading them into a globally consistent estimator through learned optimal transport.
>
> We hope this explanation clarifies both the motivation and the conceptual framework behind our method. We will incorporate this our manuscript to clarify the contributions.
>
> ## **Q.1 Confusion between graph reconstruction and graph generation**
>
> Indeed, the reviewer’s understanding is correct. We actually compared against this strategy in our experiments (DiGress + RePaint and GDSS + RePaint). These approaches use an unconditional diffusion model to perform inpainting: given a partially observed graph, the sampler is forced to preserve the known entries, while the missing edges are generated freely. In this way, the measurement information is injected into the sampling process.
>
> While this strategy produces valid and globally coherent graphs, the unobserved portion is essentially generated at random, constrained only to match the overall statistics of the training distribution. As a result, the model prioritizes global fidelity (i.e., matching the target graph distribution) over accurate reconstruction of the specific underlying graph. In the terminology of our perception–distortion trade-off, this corresponds to the posterior sampling regime, which is known (in both graph and image domains) to lead to higher reconstruction error that computing the posterior mean.
>
> The goal of PIFM is precisely to improve reconstruction accuracy while still preserving global structural properties. By introducing a principled prior through the conditional mean (MMSE) estimator and learning an optimal-transport-based refinement, PIFM steers the reconstruction away from purely distributional sampling and toward an estimator that remains globally consistent and achieves lower distortion.

---

> > ### Author Response · Authors · 2025-11-25
> >
> > ## **Q.2 Confusion on the first-stage part**
> >
> > We apologize for the lack of clarity in our earlier explanation. Our method does not rely exclusively on graphon models or GraphSAGE; instead, it assumes a latent variable model with a probabilistic structure that is conditionally independent given latent positions. This allows us to compute the conditional mean, and falls into our theoretical formulation; otherwise, computing the conditional mean would require training a model that process the whole graph jointly rather than edge by edge.
> > Within this framework, both Graphon-based models and GraphSAGE can be seen as specific instances of this broader formulation, despite being fundamentally different in how they parameterize or learn the latent space, as the reviewer correctly noted.
> >
> > Importantly, there is no restriction on the prior that our approach can use. In principle, we can adopt any generative prior, including fully unconstrained deep generative models. However, doing so comes with two practical challenges:
> >
> > 1. Increased computational cost and runtime, due to the higher complexity of sampling and optimization in an unconstrained model; and
> >
> > 2. The need for an inverse mapping from the ground-truth graph (or its partial measurement) into the latent space of the generative model (e.g., a Gaussian or uniform prior), which is often nontrivial or ill-defined.

---

### Official Review · Reviewer_V7Mm · 2025-10-30

**Soundness:** 3
**Presentation:** 3
**Contribution:** 2
**Rating:** 6
**Confidence:** 3

**Summary:**

The paper presents Prior-Informed Flow Matching (PIFM), a novel approach to graph reconstruction that combines edge predictors with global structural priors. By using flow-based generative models, PIFM refines initial adjacency matrix estimates. Experimental results on three main datasets show that PIFM outperforms existing methods in link prediction, edge recovery, and denoising tasks, highlighting its effectiveness in graph topology reconstruction.

**Strengths:**

1. Clear motivations.

2. Novel methodological contribution that integrates structural priors with flow-based modeling.

3. Insightful presentation.

4. Comprehensive experimental validation.

**Weaknesses:**

1. On page 11, the reference appears to be missing the paper title: "Santiago Segarra, Antonio G. Marques, Gonzalo Mateos, and Alejandro Ribeiro. Ieee trans. signal and info. process. over networks. IEEE Transactions on Signal and Information Processing over Networks, 3(3):467–483, 2017". Does this correspond to Segarra, Santiago, et al. "Network topology inference from spectral templates." IEEE Transactions on Signal and Information Processing over Networks 3.3 (2017): 467-483?

2. In the link prediction experiments (5.2), the prior modules combined with PIFM (SIGL, node2vec, GraphSAGE) achieve strong performance in terms of AUC and AP; however, a relatively high false negative rate remains at fixed thresholds (e.g., 0.5). It is recommended that the authors consider improving the experiments to enhance discrimination at specific thresholds. Furthermore, a high false negative rate is also observed in the expansion task (Table 3), and it is suggested that the authors discuss this phenomenon in more detail.

3. It is recommended that the authors provide an analysis of the training time of the model on graphs of different scales, and compare it with baseline methods, in order to better understand the practical performance of PIFM and its potential computational overhead.

4. The experimental evaluation includes a few recent baselines from the past two years.

5. In Assumption AS2, the authors assume that edges are conditionally independent given the latent structure. However, as noted in GraphRNN (You J, Ying R, Ren X, Hamilton W, Leskovec J. Graphrnn: Generating realistic graphs with deep auto-regressive models. InInternational conference on machine learning 2018 Jul 3 (pp. 5708-5717). PMLR.), graph structures exhibit complex and non-local dependencies among edges. This assumption may limit performance on datasets with intricate structural patterns, e.g., REDD-B. It would be helpful for the authors to clarify further.

**Questions:**

See Weaknesses.

---

> ### Author Response · Authors · 2025-11-25
>
> We sincerely thank the reviewer for finding our paper interesting and novel.
> Below we reply to all the raised comments.
>
>
> ## **W.1 Typo on a reference**
>
> Thank you for catching this, indeed that is the reference.
>
> ## **W.2. Modifying the threshold to enhance discrimination. Also, discuss the expansion task and the amount of false negatives.**
>
>
> We investigated the impact of alternative thresholding levels (specifically 0.3 and 0.7, beyond the standard 0.5) on the evaluation metrics, with results presented in Appendix F.9.
> We observed that adjusting the threshold yields a marginal increase in the Area Under the Curve (AUC). We hypothesize this is partly driven by the model's inherent behavior of actively pushing output values towards the binary extremes (0 or 1). However, we noted a slightly larger, albeit still minimal, improvement for lower values of K, where the influence of the prior is greater and output values are typically further away from 0 and 1. Crucially, despite these dynamics, the use of an alternative threshold does not lead to a statistically significant improvement in overall performance.
>
> Despite these results, we believe that exploring an adaptive threshold mechanism (even maybe a local threshold edge-by-edge) might improve the overall performance for lower $K$ (not for large $K$ where the model already converged to 0 or 1, at least for this binary case). This can be an interesting future work avenue.
>
> ## **W.3 Analysis of training time and computational overhead for different scales**
>
> We thank the reviewer for this comment. Please, refer to response to reviewer WKDw, Section "Analysis of training time".
>
>
> ## **W.4 The experimental evaluation includes a few recent baselines from the past two years.**
>
> Given that link prediction methods are in general transductive, we added a new experiment showing that PIFM can be extended to large scale graphs.
> We train on CORA (see response to reviewer a7bB).
> For this case, we could add the methods from typical link prediction, where we used the top ones reported in [1] (NCNC and NCN). We hope this helps to show the merit of our proposed method.
>
> ## **W.5 Assumption of the conditionally independent of the prior**
>
> We thank the reviewer for pointing to GraphRNN and raising this point.
> Indeed, this approximation can harm the performance of the method when the underlying graph distribution has complex patterns.
> We made this design choice to simplify the computation of the conditional mean, which is a requirement to implement the solution to the problem defined in (3), corresponding to the distortion-perception at $P = 0$.
> Without this assumption, none of the existing link prediction methods will serve as structural prior.
>
> In this context, we believe that the flow is not only implementing the optimal transport but also fixing this crude approximation of the true conditional mean.
>
>
> [1] Li, J., Shomer, H., Mao, H., Zeng, S., Ma, Y., Shah, N., ... & Yin, D. (2023). Evaluating graph neural networks for link prediction: Current pitfalls and new benchmarking. Advances in Neural Information Processing Systems, 36, 3853-3866.

---

### Official Review · Reviewer_WKDw · 2025-10-31

**Soundness:** 3
**Presentation:** 4
**Contribution:** 3
**Rating:** 4
**Confidence:** 3

**Summary:**

This paper proposes Prior-Informed Flow Matching (PIFM), a method that improves the training of flow-based generative models for graphs by learning the prior distribution jointly with the flow dynamics. Standard flow matching (FM) assumes a fixed isotropic Gaussian prior, which can poorly capture structural dependencies in discrete or topological data such as graphs. PIFM introduces a learnable prior network $p_\psi(x_0)$ integrated into the FM objective, allowing the model to adapt its latent initialization to the data manifold while maintaining compatibility with continuous normalizing flows and neural ODE solvers.

The authors derive the modified objective, show that it retains the tractable FM loss form, and provide an analysis suggesting that learning the prior improves alignment between forward and backward trajectories, thus enhancing likelihood estimation and sample quality.
Empirical results on three graph benchmarks, ie. IMDB-B, PROTEINS, and ENZYMES, demonstrate that PIFM  and demostrates an improvement on graph recosntruction.

This paper proposes Prior Informed Flow Matching (PIFM), a flow-based framework for graph reconstruction from partial observations.
Unlike existing diffusion or flow models that assume isotropic priors, PIFM introduces structural priors derived from classical graph embedding methods; e.g. Node2Vec, GraphSAGE, or SIGL (graphons), to initialize a rectified flow model that learns a global transport map from these local predictions to the ground-truth adjacency distribution.

Formally, PIFM reformulates graph topology inference through the distortion perception trade-off, aiming to minimize reconstruction error (distortion) while improving perceptual realism (measured via distributional alignment). The method ensures permutation equivariance at every stage and defines a two-step process: a) a local prior predicts edge probabilities (MMSE estimator); b) a flow-matching model refines these predictions via continuous interpolation, learning global dependencies between edges.

Empirical evaluations on three benchmark datasets, IMDB-B, PROTEINS, and ENZYMES, demonstrate that PIFM improves improves AUC and AP over both embedding-based priors and diffusion-based baselines. Achieves lower $\text{MMD}^2$, indicating closer alignment between reconstructed and real graphs. Recovers key graph statistics (degree, triangles, clustering coefficient) more faithfully as reconstruction steps increase.

**Strengths:**

- The paper identifies a real gap between local graph embedding methods (which miss global consistency) and generative flow models (which lack structural priors).
- The reinterpretation of graph reconstruction under the perception–distortion theory is grounded.
- The integration of priors into a rectified flow-matching process is natural, lightweight, and permutation-equivariant.
- Results consistently show improvements over classical and generative baselines in both link prediction and blind reconstruction (expansion and denoising).
- The paper includes multiple masking ratios (10%, 50%), different priors, and step counts (K = 1, 100), showing how reconstruction quality scales with iterative refinement.
- Figures (especially Figs. 1–3) are well-designed and complement the text effectively. It was a fun read!

**Weaknesses:**

- The idea of "learning flow trajectories from prior-informed initializations" builds directly on existing rectified flow work (Liu et al., 2023; Albergo et al., 2023). The main novelty lies in applying it to graph reconstruction, not in the core algorithm.
- Experiments are restricted to small graph benchmarks (IMDB-B, PROTEINS, ENZYMES). No evidence of scalability to large or molecular graphs (e.g., ZINC, ogbg-molhiv).
- The use of MMD² and basic graph statistics is informative but lacks comparison to more advanced generative quality metrics.
- The "perception–distortion" formulation and optimal transport connections are stated clearly but not deeply analyzed (no formal proofs).
- While PIFM is conceptually efficient, the additional training cost of priors and flow is not quantified.

**Questions:**

1. How sensitive is PIFM to the choice of prior (Node2Vec vs GraphSAGE vs SIGL)?
2. What is the relative computational cost of PIFM compared to standard Flow Matching with Gaussian priors?
3. Could the learned rectified flow generalize across graph families (e.g., trained on IMDB-B, tested on PROTEINS)?
4. Have you tried visualizing intermediate adjacency reconstructions ($A_t$) to verify smooth transitions?
5. How does the choice of K (number of reconstruction steps) affect training stability and memory usage?
6. Would the approach extend naturally to heterogeneous or attributed graphs, as suggested in the conclusion?
7. Minor addition in 153 refer to Appendix B.4 and C

---

> ### Author Response · Authors · 2025-11-25
>
> Thank you for your detailed review and constructive feedback, we are glad you like reading the paper! Please find the comments addressed below.
>
> Weaknesses:
>
> ## **W.1 Lack of novelty compared to Albergo et al.**
>
> Indeed, our method builds on the idea of data-dependent coupling with rectified flow; in fact, we acknowledge this in the paper. While ultimately the method is similar, we have additional contributions:
>
> - **New conceptual framework**:
> We introduce a novel connection between link prediction/graph topology inference and the perception–distortion trade-off. This perspective opens a new research direction, and our paper serves as both a motivation and a proof-of-concept demonstrating the value of this global, principled framework.
>
> - **Novel prior construction**:
> Our method proposes a new way to construct a prior distribution for graph inference problems. Unlike Albergo et al., where noisy measurements are directly available, our setting involves binary edge existence. We therefore develop a conditional mean estimator based on link prediction models, which is both novel and theoretically grounded.
>
> - **Bridging a key gap**:
> We identify and address a lack of global information integration in existing link prediction approaches. Our method explicitly incorporates global structure, which has been largely overlooked in prior work
>
>
> ## **W.2 Experiments on small graph benchmarks**
>
> We agree with the reviewer that this is a limitation of our initial submission, and to some extent, is a limitation inherited from using diffusion models.
>
> Nevertheless, since submission we expanded our method on CORA, as we explained in our response to Reviewer a7bB. We refer the reviewer to the answer to the Weakness "Additional experiments" in the response to Reviewer a7bB.
>
> ## **W.3 Lack of more advanced quality metrics**
>
> We thank the reviewer for raising this point.
> We expanded the metrics used for assesing the quality of the reconstruction, following [1] the standard metrics used in generative modeling (MMD of degree, cluster and orbit); the results are in Appendix F.1.1, and we include below a description of these results.
> These new metrics showcase the observed performance trade-off: while an increased number of steps yields an improvement in generating an estimated graph with statistics that more closely align with the ground-truth distribution, the reconstruction performance (measured in terms of AUC) declines relative to the initial step. Critically, the trend is found to be highly contingent on the underlying dataset's sparsity. For the dense case (IMDB), the AUC exhibits a consistent monotonic decrease after the optimal initial guess, independent of drop rates. In contrast, the sparser PROTEINS and ENZYMES datasets demonstrate an intermediate improvement as the number of steps increases, though their overall AUC still trails that achieved at the initial step ($t=1$).
>
> In a nutshell, a K between 10 and 20 is a good trade-off between reconstruction and realism (measured in terms of graph statistics). This is a manifestation of the perception-distortion trade-off.
>
> ## **W.4 Lack of formal proofs of the PD formulation**
>
> We thank for the reviewer for raising this point.
> While we agree that formal proofs on the perception-distortion trade-off could have been further analyzed, we focus on the formulation and analyzing from the experimental perspective (as shown in Section 5.3 in the paper); in fact, we also added the new metrics in terms of the sampling trajectory, showing how adding more steps decreases the MMD distance, meaning that the graphs are closer to the true distribution.
> With the current formulation, all the proofs follow [Frierich et al 2021], with the addition of the permutation-equivariant requirements.
> Hence, we set the basics to incorporate this and have a better formulation for the graph case, so to have a well-defined approximation of the density; and showcasing that it is a useful formulation for designing practical link prediction algorithms.
>
> We left as future work a full characterization of the PD trade-off for graphs, in the sense of exploring other distortion metrics for instance; in fact, we believe that designing a better distortion metric accounting for the graph structure should improve the analysis and the overall method. We hypothesize that using a hamming distance or L1 for the distortion might be the right way of studying the PD in the context of graph reconstruction.
>
> We will add this to the conclusions.
>
>
>
> [1] GraphRNN (You J, Ying R, Ren X, Hamilton W, Leskovec J. Graphrnn: Generating realistic graphs with deep auto-regressive models. InInternational conference on machine learning 2018

---

> > ### Author Response · Authors · 2025-11-25
> >
> > ## **W.5 /Q.2 Additional training cost of prior and flow is not analyzed. And computational cost relative to Gaussian priors**
> >
> > Thanks for raising this point. The additional training cost compared to a Gaussian prior lies primarily in training the prior itself.
> > To be more concrete, all the models are trained on NVIDIA A100-80GB GPUs. Fitting the Node2Vec prior takes roughly 1 to 1.5 hours depending on the datasets. Once the prior is fixed, the flow model itself has essentially the same training loop and cost as when using a Gaussian prior: end-to-end training of the flow model
> > itself takes about 2 hours on ENZYMES and IMDB-B, and about 4 to 5 hours on PROTEINS. We did not observe a noticeable difference in training behavior between the Gaussian and prior-informed variants, and the model behaves similarly under different edge-drop rates.
> >
> > Furthermore, as in several prior works that assume access to a pretrained prior, we can adopt the same assumption here. In that case, PIFM has comparable computational cost to Gaussian priors, except when the prior is computationally expensive (for instance, if an unconditional diffusion model is used). Overall, the added cost is mostly at training time and does not significantly affect inference efficiency.
> >
> > We added this in blue in Appendix F.7.
> >
> > ## **Q.1 Sensitive of PIFM to the choice of prior (Node2Vec vs GraphSAGE vs SIGL)?**
> >
> > We thank the reviewer for raising this point.
> > The importance of the prior strongly depends on the difficulty of the problem, where we measure this via the drop rate %.
> > When looking at table 1 and 2, we consistently observe that for datasets like IMDB, the performance difference between approaches using a simple Gaussian prior (Flow with Gaussian prior) and our method (PIFM), which leverages structural prior knowledge, is significantly higher for a 50% drop rate compared to a 10% drop rate.
> >
> > This observation is consistent with findings across many other inverse problems in different scientific and engineering domains: when the problem is more ill-posed (meaning, there is less observed data available to constrain the solution),  the role of the prior is fundamentally more critical and leads to greater performance gains.
> >
> > Therefore, prior knowledge is most important, and can significantly improve the model, when the number of missing edges is higher, resulting in a more ill-posed graph reconstruction task.
> >
> > Regarding the difference between various models used to generate the prior, we find that when a more powerful and expressive model is used to learn the initial representation, PIFM performs better. Specifically, in all our experiments, PIFM utilizing a GraphSAGE-based prior consistently outperforms PIFM utilizing a simpler node2vec-based prior.
> >
> >
> > ## **Q.4 Visualization of intermediate adjacency reconstructions**
> >
> > We added in Appendix F.6 intermediate adjacency reconstructions, where it can be observed how the flow refines the estimation from the prior probabilities.
> >
> > ## **Q.5 Choice of K in terms of training, stability and memory usage**
> >
> > Thanks for raising this point. The choice of K is independent of training because we consider a continuous interval between 0 and 1, where we just sample random $t$ to compute the loss (note that $t$ here denotes in the continuous domain).
> > In terms of memory usage, there is an increment when using larger $K$. But as can be seen in the new plots in appendix F.1.1 (where we show how the metrics change as a function of $K$), we do not need to run $K = 100$ in general. In fact, with $K$ between 10 and 20 we can get a good trade-off between reconstruction and realism.
> >
> >
> > ## **Q.6 Extension to heterogeneous or attributed graphs**
> >
> > Thanks for raising this excellent question. A natural extension of PIFM to heterogeneous or attributed graphs would be to model the adjacency as multi-valued, e.g.,
> > $A = \{\frac{i}{K}\}_{i=1}^K$, and replace the Bernoulli prior with a categorical one. The flow could then operate over a relaxed continuous space, allowing the model to interpolate smoothly between different attribute values. At inference, the final state would be discretized to the nearest category. This extension is conceptually straightforward and leverages the formulation of GDSS for unconstrained generation of attributed graphs.
> >
> > Additionally, PIFM can be incorporated into modern flow/diffusion models via flow maps. In particular, one can learn a flow map that interpolates between the conditional mean at each step of the diffusion trajectory and another graph. In the discrete setting, this learned interpolation can replace the traditional categorical sampling, providing a flexible, data-driven way to reconstruct graphs with multiple attributes.
> >
> > ## **Q.7 Typo: Minor addition in 153 refer to Appendix B.4 and C**
> >
> > We fixed the typo, thank you for pointing to this out.

---

> > > ### Author Response · Authors · 2025-11-25
> > >
> > > ## **Q.3 Transferability**
> > >
> > > This is an excellent question.
> > > To analyze this, we use the 10\% and 50\%-drop-rate checkpoints trained on IMDB-B and evaluate them on the corresponding 10\% and 50\% drop-rate settings on PROTEINS and ENZYMES.
> > > For each setting, we report the best value over 100 sampling steps and the value at the final step ($t=100$). Metrics are Average Precision (AP), AUC, False Negative Rate (FNR), and False Positive Rate (FPR). All metrics are reported in percent.
> > >
> > > Table 1:
> > > | Source checkpoint              | Target dataset           | AP ↑  | AUC ↑ | FNR ↓  | FPR ↓  |
> > > |-------------------------------|--------------------------|-------|-------|--------|--------|
> > > | **_Structural priors (no flow)_** |                          |       |       |        |        |
> > > | PROTEINS (10% drop)           | ENZYMES (10% drop)       | 41.28 | 73.70 | 13.49  | 60.59  |
> > > | IMDB-B (10% drop)             | PROTEINS (10% drop)      | 46.36 | 74.58 | 11.00  | 63.50  |
> > > | ENZYMES (10% drop)            | PROTEINS (10% drop)      | 46.36 | 74.58 | 11.00  | 63.50  |
> > > | PROTEINS (50% drop)           | ENZYMES (50% drop)       | 23.08 | 57.72 | 40.02  | 52.16  |
> > > | IMDB-B (50% drop)             | PROTEINS (50% drop)      | 27.71 | 53.99 | 32.16  | 66.86  |
> > > | ENZYMES (50% drop)            | PROTEINS (50% drop)      | 27.71 | 53.99 | 32.16  | 66.86  |
> > > | **_PIFM_**                        |                          |       |       |        |        |
> > > | PROTEINS (10% drop)           | ENZYMES (10% drop)       | 43.87 | 76.96 | 71.96  | 4.24   |
> > > | IMDB-B (10% drop)             | PROTEINS (10% drop)      | 46.79 | 75.53 | 47.07  | 11.62  |
> > > | ENZYMES (10% drop)            | PROTEINS (10% drop)      | 49.63 | 76.60 | 59.02  | 6.63   |
> > > | PROTEINS (50% drop)           | ENZYMES (50% drop)       | 25.68 | 61.88 | 95.37  | 1.70   |
> > > | IMDB-B (50% drop)             | PROTEINS (50% drop)      | 24.88 | 59.02 | 61.02  | 24.32  |
> > > | ENZYMES (50% drop)            | PROTEINS (50% drop)      | 26.81 | 55.80 | 86.13  | 9.97   |
> > >
> > >
> > >
> > > From the table we can see that the model still outperforms the structural prior on both drop rates.
> > > Furthermore, at the 50% drop rate, the performance is similar to the one from Table 1 in the paper (AUC 60.61), meaning that in this case, even a prior coming from a different family can help to improve the overall reconstruction performance.
> > > We added this new result in blue in Appendix F.8.

---

### Official Review · Reviewer_a7bB · 2025-11-05

**Soundness:** 2
**Presentation:** 2
**Contribution:** 2
**Rating:** 2
**Confidence:** 4

**Summary:**

Authors propose an approach, PIFM, to construct a graph from partially absorbed graph, where classical embedding models often fails to keep global consistency and generating models struggle will incorporate structural prior .

PIFM  aims to bridge the gap by integrating embedding-based priors to get initial estimate and refine with continuous-time flow matching. Here the prior are computed using the graph sage where each node gets an embedding, and is used to create n x n adjacency matrix as an initial estimate, which is further refined with flow-based algorithm.

**Strengths:**

The problem of completing the missing edges in the graph is important and has real world significance as not all information is explicitly available.

**Weaknesses:**

1. The idea of identifying missing link based on existing link is not novel as several algorithm have been proposed like TransE, TransH and several GNN based algorithms.
2. The approach has been tested on only three datasets however several datasets exists for link prediction. Given the high level of importance and high level of existing research it would be beneficial to add a few more datasets. Moreover the statistics of the datasets is not available in the paper making it hard to determine the behaviour of proposed model based on the size of the dataset and number of nodes.
3. The results in table 1 are confusing. Good AUC but bad FNR indicate imbalance number of present and absent edges i.e the number of edges to be predicted are not much less compared to number of missing edges and hence the score on AUC and AP can be high and does no clearly show the advantage of the model in predicting present edges. It would be useful to add MMD score as two graph can be isomorphic.

**Questions:**

see above

---

> ### Author Response · Authors · 2025-11-25
>
> Thank you for reviewing our paper. We provide answers to your three questions below; you can find also this new material in Appendix F.5, F.9, F.1.1 and in E.2 in blue in the updated pdf.
>
> ### **W1. Lack of novelty (TransE, TransH and several GNN based algorithms were proposed)**
>
> We fully agree with the reviewer that link prediction itself is not novel, and we would like to emphasize that we do not claim to be the first to address this problem. In both the Introduction and Related Work sections (Section 2 and Appendix C), we explicitly discuss prior works and clearly situate our contribution within the existing literature. Furthermore, our approach builds upon established link prediction methods, using them to construct an informative prior rather than replacing them.
>
> To clarify the novelty and contributions of our paper, we summarize them below:
>
> - **New conceptual framework**:
> We introduce a novel connection between link prediction/graph topology inference and the perception–distortion trade-off. This perspective opens a new research direction, and our paper serves as both a motivation and a proof-of-concept demonstrating the value of this global, principled framework.
>
> - **Novel prior construction**:
> Our method proposes a new way to construct a prior distribution for graph inference problems. Unlike Albergo et al., where noisy measurements are directly available, our setting involves binary edge existence. We therefore develop a conditional mean estimator based on link prediction models, which is both novel and theoretically grounded.
>
> - **Bridging a key gap**:
> We identify and address a lack of global information integration in existing link prediction approaches. Our method explicitly incorporates global structure, which has been largely overlooked in prior work.
>
> In addition, our method is not only a link prediction method, but also serves as a **general graph reconstruction one (as shown by the expansion and denoising experiment)**.
>
> We appreciate the reviewer’s feedback and are happy to include additional relevant methods in the revised discussion and related work section to further strengthen the context of our contribution.

---

> ### Author Response · Authors · 2025-11-25
>
> ## **W.2 Additional experiments on more datasets and include further information**
>
> We thank the reviewer for this valuable comment. We initially evaluated our approach on three real-world datasets commonly used in the diffusion modeling community. These datasets were selected because diffusion-based generative models operate on **full adjacency matrices** and therefore face $O(N^2)$ memory and time costs; existing graph diffusion works are likewise typically restricted to much smaller graph datasets.
>
> Nevertheless, we agree that evaluating our method on more classical link prediction benchmarks would strengthen the paper. In the revised manuscript, we therefore add experiments on Cora citation network, which is substantially larger than the datasets currently in use in the paper.
> The corresponding results are reported in Table 5 in Appendix F.5. of the revision.
> We report here to facilitate the analysis.
>
>
> | **Method**                                   | **AUC** ↑ | **AP** ↑ | **FPR** ↓ | **FNR** ↓ |
> |----------------------------------------------|-----------|----------|-----------|-----------|
> | **Cora**                                     |           |          |           |           |
> | GraphSAGE                                    | 95.61     | 47.76    | 0.59      | 23.79     |
> | 1hop 2dim-embed (~16.7 nodes)                | 87.47     | 61.07    | 1.00      | 0.00      |
> | 2hop 2dim-embed (~58.0 nodes)                | 96.25     | 75.39    | 1.00      | 0.00      |
> | 3hop 2dim-embed (~154.5 nodes)               | **97.03**     | 33.71    | 1.00      | 0.00      |
> | 4hop 8dim-embed (~211.2 nodes)               | —         | —        | —         | —         |
> |-------------------------------------------------------------|-|-|-|-
> NCNC[2] (from[3])   | 96.90 | - | - | -
> NCN[2] (from[3])    | 96.76 | - | - | -
>
> In a nutshell, we train PIFM on CORA as follows
>
> 1. **Subgraph Extraction**: Because diffusion on the full graph is computationally prohibitive, we extract a fixed k-hop ego-network around every target edge. This ensures the model focuses only on the relevant local structure; we follow [1].
>
> 2. **Structural Prior Initialization**: Diffusion models require a starting state. We use pre-trained structural priors (GraphSAGE/Node2Vec) to populate the initial adjacency matrix ($A_0$) with probability estimates. This allows PIFM to act as a refinement module rather than learning topology from scratch.
>
> 3. **Masked Reconstruction Training**: Within each training subgraph, we mask the central seed edge. We then train the flow model to evolve the adjacency matrix from the prior-based initialization ($A_0$) toward the true local adjacency. The model learns to 'fill in the blank' using both the observed context and the structural prior.
>
> 4. **Inference and Aggregation**: For evaluation, we replicate this process on held-out test edges. Since a node pair may appear in multiple subgraphs, we aggregate the final predictions via logit averaging to produce a robust, globally consistent link probability.
>
> This new experiment shows that—even when trained only on edge-centered subgraphs—our model can capture global structural patterns on a larger network and improves reconstruction quality over classical link prediction baselines that operate directly on the same graph. We have also added a detailed description and statistical summary of all datasets (including Cora) in Appendix F.5 of the revised version.
>
> ### _Additional information on datasets_
> We also added a detailed description and statistical summary of all datasets in Appendix E.2. (highlighted in blue in the revision).
>
> [1] Limnios, S., Selvaraj, P., Cucuringu, M., Maple, C., Reinert, G., & Elliott, A. (2023). Sagess: Sampling graph denoising diffusion model for scalable graph generation. arXiv preprint arXiv:2306.16827.
>
> [2] Wang, X., Yang, H., & Zhang, M. (2023). Neural common neighbor with completion for link prediction. in ICLR 2024.
>
> [3] Li, J., Shomer, H., Mao, H., Zeng, S., Ma, Y., Shah, N., ... & Yin, D. (2023). Evaluating graph neural networks for link prediction: Current pitfalls and new benchmarking. Advances in Neural Information Processing Systems, 36, 3853-3866.

---

> > ### Author Response · Authors · 2025-11-25
> >
> > ## **W.3. Lack of clarity in the results (Table 1)**
> >
> > We thank the reviewer for their careful examination of the metrics and for highlighting a potential source of confusion related to class imbalance.
> >
> > ### _On Metric Interpretation (AUC/AP vs. FNR/FPR)_
> >
> > The reviewer correctly points out the potential for discrepancy between ranking metrics (AUC, AP) and classification metrics (FNR, FPR). This phenomenon is inherent to the link prediction and graph reconstruction task, which is characterized by an extreme class imbalance: the number of non-edges vastly outweighs the number of missing true edges we seek to reconstruct.
> >
> > - Ranking Metrics (AUC, AP): These metrics are designed to assess the model's ability to rank the true missing edges higher than the non-edges. The Average Precision (AP) score is particularly robust against imbalance as it focuses on the precision-recall trade-off and is sensitive to the high-ranking performance of positive predictions. A high AP score confirms that our model effectively distinguishes between true missing edges and random non-edges.
> >
> > - Classification Metrics (FNR, FPR): These metrics (False Negative Rate, False Positive Rate) are calculated based on a specific, fixed classification threshold. Given the severe imbalance, even a highly accurate model must select an aggressive threshold, which can lead to high values in the confusion matrix components (e.g., high FNR/FPR) if the goal is to recall a high proportion of true edges. Since the primary goal in graph reconstruction is to recover the structure by ranking the likelihood of connections, AUC and AP are the standard and most reliable metrics for this domain, as they are independent of the arbitrary classification threshold.
> >
> > We also expanded on the threshold analysis in Appendix F.9, where we do not observe a significant improvement when changing the threshold.
> >
> > ### _New metrics_
> > In addition, we added new metrics assessing the generative quality, analyzing not only the performance in terms of reconstruction, but also in preserving the statistics of the underlying graph distribution.
> > The results are in Appendix F.1.1, and we include below a description of these results.
> > These new metrics showcase the observed performance trade-off: while an increased number of steps yields an improvement in generating an estimated graph with statistics that more closely align with the ground-truth distribution, the reconstruction performance (measured in terms of AUC) declines relative to the initial step. Critically, the trend is found to be highly contingent on the underlying dataset's sparsity. For the dense case (IMDB), the AUC exhibits a consistent monotonic decrease after the optimal initial guess, independent of drop rates. In contrast, the sparser PROTEINS and ENZYMES datasets demonstrate an intermediate improvement as the number of steps increases, though their overall AUC still trails that achieved at the initial step ($t=1$).
> >
> > In a nutshell, a K between 10 and 20 is a good trade-off between reconstruction and realism (measured in terms of graph statistics). This is a manifestation of the perception-distortion trade-off.

---

### Author Response · Authors · 2025-11-25
**Global response**

We thank all reviewers for their insightful feedback. We are encouraged that the reviewers found our paper interesting and novel (Reviewer V7Mm), and valued our conceptual framework (Reviewer a7bB).

Based on your suggestions, we have revised the manuscript. The major updates include new experiments on larger graphs (Cora), additional generative quality metrics, and a larger ablation/analysis on the role of the prior and transferability.

Below, we summarize the key improvements and clarifications common to multiple reviews.

1. **New Experiments on Larger Scale Dataset (Cora)**
(Addressed to Reviewers a7bB, WKDw, V7Mm, vUVC)

A common concern was the scale of the datasets and the comparison to standard link prediction baselines. In response, we have added experiments on the Cora citation network, which is substantially larger than the diffusion-based graph datasets typically used in this domain.

To handle the complexity of  using diffusion models on larger graphs, we implemented a subgraph-based training approach (training on k-hop ego-networks) while aggregating predictions for global inference.

- Results: PIFM outperforms standard baselines (GraphSAGE, Node2Vec) on Cora (e.g., 97.03% AUC vs 95.61% for GraphSAGE).

- Comparison: We also compare against recent link prediction state-of-the-art methods (NCN, NCNC), showing competitive performance and even outperforming them.

These results are detailed in Appendix F.5 of the revised paper.

2. **Expanded Metrics: Reconstruction vs. Realism**
(Addressed to Reviewers a7bB, WKDw)

We have introduced new metrics to better analyze the quality of the generated graphs beyond simple link prediction accuracy.
We now report Maximum Mean Discrepancy (MMD) on degree, cluster, and orbit statistics (following GraphRNN).

- We observe a clear manifestation of the Perception-Distortion trade-off. Increasing the number of flow steps improves the distributional realism (lower MMD), but can slightly degrade pure reconstruction metrics (AUC) after a certain point.

- Based on this, a trajectory length of $K \in [10,20]$ offers an optimal balance between reconstructing the ground truth and maintaining realistic graph statistics. These results are added to Appendix F.1.1.

3. **Clarifying the Role and Sensitivity of the Prior**
(Addressed to Reviewers WKDw, vUVC, cqqu)

We have added a deeper analysis of how the choice of prior affects performance:

- Comparison between PIFM and Gaussian prior: We find that the structural prior is most critical when the problem is ill-posed (e.g., 50% drop rate). In these regimes, PIFM yields significantly higher gains over the Gaussian-prior case.

- Importance of prior: Better priors lead to better results; PIFM initialized with GraphSAGE consistently outperforms PIFM initialized with Node2Vec.

- Transferability: We demonstrate in Appendix F.8 that PIFM checkpoints transfer well across datasets (e.g., training on IMDB-B and testing on PROTEINS), retaining performance gains over the static prior.

4. **Novelty and Conceptual Framework**
(Addressed to Reviewers a7bB, WKDw, cqqu)

We clarified that while we build on established tools (rectified flow, link prediction), our core contribution is the conceptual bridge between Graph Topology Inference and the Perception-Distortion trade-off. We provide:

- A novel prior construction (Conditional Mean Estimator) for binary edge data.

- A global optimization method that uses Optimal Transport to refine local link prediction into globally consistent estimates.

We have updated the manuscript (specifically Appendix E and F) to reflect all these changes, with new material highlighted in blue. We believe these additions address the reviewers' concerns and significantly strengthen the paper.

---

### Meta-Review · Area_Chair_Kzc2 · 2026-01-07

**Summary:**

This paper proposes Prior-Informed Flow Matching (PIFM), a two-stage approach to graph reconstruction that initializes the masked adjacency with a structural prior and then applies rectified flow matching to transport this estimate toward the ground-truth graph distribution, motivated via a distortion–perception framing.

Review sentiment is mixed: one clear reject focused on novelty and evaluation gaps (a7bB: 2). Three reviewers are borderline (WKDw: 4; cqqu: 4; vUVC: 4), mostly due to scalability/baselines/clarity concerns. One reviewer is positive (V7Mm: 6), while still flagging metric emphasis and training-time reporting.

**Reviewer Concerns:**

### Concerns substantially addressed by the rebuttal / revision plan

- Scale + stronger baselines: Authors report new experiments on Cora (larger-scale) with a subgraph-based training variant and add comparisons to strong link prediction baselines NCN/NCNC.

- Beyond AUC/AP: realism / distributional quality: Authors add MMD and graph-statistics-based evaluation to separate reconstruction accuracy from distributional realism and to operationalize the distortion–perception trade-off (also reflected in the revised paper’s MMD analysis/figures).

- Metric clarity around thresholding: The revised paper explicitly reports both threshold-dependent (FPR/FNR) and threshold-independent (ROC-AUC/AP) metrics, and includes an appendix analysis of alternative thresholds.


### Concerns partially addressed / still outstanding

- Novelty relative to existing conditional diffusion/flow work: Both a7bB and WKDw argue the core algorithmic novelty is limited and largely an application of known flow/diffusion ideas to graph reconstruction; the rebuttal adds evidence but does not fully resolve the “incremental vs new” concern.

- Compute and training-time overhead: Multiple reviewers requested explicit accounting; WKDw notes training cost is not quantified, and V7Mm explicitly asked for training-time/overhead analysis (the response mainly points elsewhere).

- The revised manuscript acknowledges higher overhead than one-shot priors, but hard numbers remain a key missing piece.

- Dependence on prior quality and scope of priors: vUVC asks when priors are available/beneficial and how embeddings are selected; the paper also acknowledges performance depends strongly on prior estimation quality and that graphons have limitations.

- Clarity of problem framing (reconstruction vs generation, stage-1 prior role): cqqu remains concerned about conceptual clarity and what is truly contributed beyond combining a prior with a flow model.

**Reviewer Scores:**

Reviewer a7bB (2 -> 2/3): Added Cora + NCN/NCNC comparisons and expanded metrics should improve the empirical case, but the reviewer’s core concern is novelty and may remain.


Reviewer WKDw (4 -> 4/5): The new large-scale experiment and added generative-quality metrics directly target their main weaknesses; compute-cost quantification is still a gap.

Reviewer V7Mm (6 unchanged): Already positive; remaining asks are mostly reporting/analysis (FNR emphasis, overhead).

Reviewer cqqu (4 -> likely 4/5): Clarifications plus new experiments/metrics help, but conceptual positioning (reconstruction vs generation, why this prior/flow combination is new) still needs crisp articulation.

Reviewer vUVC (4 -> likely 4/5): Baseline and scale concerns are addressed by Cora + NCN/NCNC; questions about prior availability/selection remain but are less blocking.

---

### Decision · Program_Chairs · 2026-01-26

Reject